# Privacy-Preserving and Fairness-Aware Federated Learning for Critical Infrastructure Protection and Resilience

## ABSTRACT

The energy industry is undergoing significant transformations as it strives to achieve net-zero emissions and future-proof its infrastructure, where every participant in the power grid has the potential to both consume and produce energy resources. Federated learning – which enables multiple participants to collaboratively train a model without aggregating the training data – becomes a viable technology. However, the global model parameters that have to be shared for optimization are still susceptible to training data leakage. In this work, we propose Confined Gradient Descent (CGD) that enhances the privacy of federated learning by eliminating the sharing of global model parameters. CGD exploits the fact that a gradient descent optimization can start with a set of discrete points and converges to another set in the neighborhood of the global minimum of the objective function. As such, each participant can independently initiate its own private global model (referred to as the *confined model*), and collaboratively learn it towards the optimum. The updates to their own models are worked out in a secure collaborative way during the training process. In such a manner, CGD retains the ability of learning from distributed data but greatly diminishes information sharing. Such a strategy also allows the proprietary confined models to adapt to the heterogeneity in federated learning, providing inherent benefits of fairness. We theoretically and empirically demonstrate that decentralized CGD (*i*) provides a stronger differential privacy (DP) protection; (*ii*) is robust against the state-of-the-art poisoning privacy attacks; (*iii*) results in bounded fairness guarantee among participants; and (*iv*) provides high test accuracy (comparable with centralized learning) with a bounded convergence rate over four real-world datasets.

## 1 INTRODUCTION

Federated learning (FL) enables participants to jointly train a global model without the necessity of sharing their datasets [33, 34, 67], demonstrating the potential promise of communication efficiency and protecting proprietary user data. It has been incorporated by popular machine learning frameworks such as TensorFlow [1] and PyTorch [51], and the burgeoning development of FL variance renders it promising to be deployed across various industries [14, 19, 37, 40, 42, 52, 64], including critical infrastructure such as energy sector [14, 19, 37, 40, 42, 52, 64]. With the widespread adoption of distributed renewable energy resources such as electric cars, battery storage, wind turbines, and solar panels, managing the distributed network through leveraging FL technology has become a complex challenge. Every participant in the network now has the potential to both consume and produce energy resources, making it crucial to maintain security and privacy, equitability, and resilience in the distributed network.

**Challenges.** Standard FL approaches, however, have been shown vulnerable to privacy leakage which stems from its parallelization of the gradient descent optimization [46, 50, 72]. During the

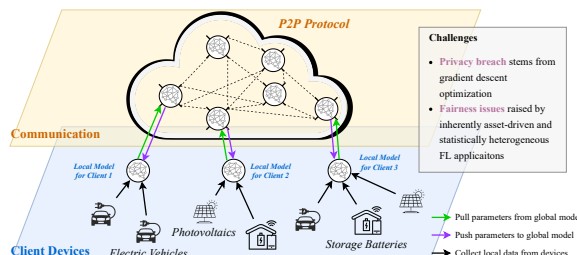

**Figure 1: An example of federated learning application in the context of critical infrastructure.**

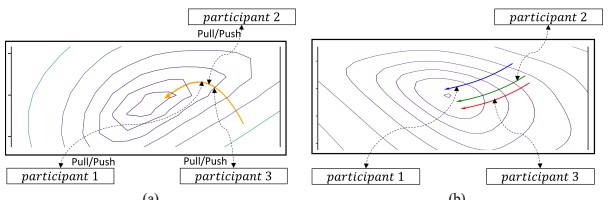

(a) Gradient descent in traditional federated learning in which participants jointly work on the same global model; (b) Confined Gradient Descent. Each participant strictly confines their own global models throughout the learning process. When the optimization algorithms converge, they reach the neighbourhood of the centralized model that was to learn centrally on the gathered data.

**Figure 2: The optimization process of the traditional FL and CGD on MNIST [36]** (both with three participants for illustration)

training process, each participant obtains the current global model parameters, works out a local gradient based on the local data, and disseminates it for others to update the global model. This paradigm guarantees data locality, but the shared local models/gradients imply sensitive attributes of local training samples, as revealed by recent studies [20, 59]. In addition, many FL applications are inherently asset-driven and statistically heterogeneous, resulting in fairness challenges. For example, without reconciliation of privacy and fairness in the context of sustainable critical infrastructure, *i.e.,* peer-to-peer (P2P) energy network (see Figure 1), privileged individuals, groups and organizations can have advantageous access to the power grid and benefit at the cost of underserved groups, substantiated by the fact that merely implementing differential privacy techniques may lead to disproportionate impact on minority communities [3, 55].

**Solution.** To ensure private and equitable participation of community members and responsible use of their data in providing secure, reliable, and sustainable energy services through FL, this work aims to enhance the privacy preservation with bounded fairness guarantee in FL by *eliminating the explicit sharing of the centralized global model* to decrease the dependency among participants. We therefore propose a new optimization algorithm named Confined Gradient Descent (CGD) that enables each participant to learn its own private "global" model. The CGD participants initiate their own global models locally, and then strictly confine these models

within themselves throughout the whole training process. The updates to their own models are operated in a secure collaborative way during the training process. In such a manner, CGD retains the merit of *federation* but greatly diminishes information leakage. In the remaining of this paper, we refer to these localized and private global models as *confined models*, to distinguish them from the global model in traditional FL. In addition, CGD guarantees that each confined model, when CGD converges, is at the neighborhood of the centralized model. This property, in principle, retains desirable model accuracy. Furthermore, this strategy allows the proprietary confined models to further adapt to the heterogeneity in locality, which provides inherent benefits of fairness.

**Observation and Motivation.** CGD is inspired by an observation on the surface of the typical cost function. The steepness of the first derivative decreases slower when approaching the minimum of the function, due to the small values in the Hessian (*i.e.,* the second derivative) near the optimum [9]. This gives the function, when plotted, a flat valley bottom. As such, a gradient descent algorithm $\mathcal{A}$, when applied on an objective function $F$, could start with a set of discrete points. Iteratively descending the set using their joint gradient would lead $\mathcal{A}$ to the neighborhood of $F$'s minimum in the "flat valley bottom". Those points would also end up with similar losses that are close to the loss of the minimum.

From a high-level view,[1] we visualize in Figure 2 a comparison between the optimization process of CGD and that of a gradient decent in traditional FL on the MNIST dataset [36]. In traditional FL (Figure 2(a), every participant updates the *same* global model using their local gradients, while in CGD (Figure 2(b)), each participant learns its own confined model and hides its confined model from each other throughout the training process. CGD allows its participants to choose their own randomization strategy to initialize their confined models. For example, they can decide their own interval range of initial weights. In every training iteration, each participant independently computes the local gradient from its current confined model and local data, and then securely works out aggregated gradients so as to update their confined models. The optimization process of CGD endows FL with the capability of *advantageous privacy-utility tradeoff* and *bounded fairness* among participants.

**Contributions.** We summarize our main contributions as follows.

- **Confined Gradient Descent for Privacy-enhancing Federated Learning.** We propose a new optimization algorithm CGD for privacy-preserving decentralized FL. CGD eliminates the explicit sharing of the global model and lets each participant learn a proprietary confined model. Therefore, it can easily accommodate any FL scheme regardless of their underlying machine/deep learning algorithms. It also eliminates the necessity of a central coordinating server, such that the optimization can be conducted in a fully decentralized manner.
- **Enhanced Privacy Preservation over Traditional FL.** CGD further enhances the privacy in FL. First, we prove that CGD provides a stronger privacy guarantee (*i.e.,* in CGD, less noise is required) than the existing differentially private federated learning frameworks [2, 63]. We also experimentally demonstrate the

[1]For the visualization purpose, we follow the techniques proposed in [26, 38] to plot 2D loss landscape, including using principal component analysis (PCA) to get two principle directions as projection axis, and filter normalization to mitigate the loss value drifting caused by norm deviation.

**Table 1: A summary of CGD and related privacy-preserving FL methods.**

| Techniques | Methodology | | Desired Properties | | | | | | |
|---|---|---|---|---|---|---|---|---|---|
| | Local Model Heterogeneity | Bounded Loss (Local Model) | Advantageous Privacy-Utility Tradeoff | Robustness against Passive MIA | Robustness against Active MIA | Convergence Guarantee | Bounded Fairness | Bounded Asynchrony |
| Plain FL [10, 54] | ○ | ○ | ○ | ○ | ○ | ● | ○ | ○ |
| FL via Secure Aggregation [8, 43, 44, 48, 57, 69] | ○ | ○ | ◑ | ○ | ○ | ● | ○ | ○ |
| FL via Differential Privacy [2, 16, 29, 58, 61, 70] | ○ | ○ | ◑ | ● | ○ | ◑[1] | ○ | ○ |
| CGD (Ours) | ● | ● | ● | ● | ● | ● | ● | ● |

●: The corresponding item is supported; ◑: The corresponding item is partially supported; ○: The corresponding item is not supported.
[1] Additive noise (in most cases) affects the model accuracy [16].

advantages of CGD. For example, given test accuracy around 95% on the MNIST dataset, CGD achieves $(0.41, 10^{-5})$-DP, which is tighter than the existing methods reaching the same accuracy, *e.g.,* $(0.95, 10^{-5})$-DP in Bayesian DP [63], and $(2.0, 10^{-5})$-DP in Abadi *et al.* [2]. In addition, we also demonstrate CGD's robustness against the state-of-the-art poisoning membership inference attacks in FL [46, 50, 68]. Such attacks typically aim to infer the training samples' membership by injecting malicious attributions against the victim samples. CGD enables each local participant to examine its confined model's loss value and test accuracy with respect to its training samples. By doing this, the participant can effectively detect suspicious privacy attack against its training samples and thus determine whether to set the new iterate using the current updates.

- **Bounded Fairness.** CGD enables a more fair solution for each participant through heterogeneous confined model and bounded asynchronous optimization, *i.e.,* participants are off synchrony up to a certain bounded delay. We first provide a proof to CGD fairness-convergence theorem to demonstrate that the distance between any of the confined models and the centralized model is bounded when CGD converges. Furthermore, we empirically show that CGD decreases the variance of the test accuracy across participants compared to the state-of-the-art fairness-aware FL [39].
- **Functional Evaluations.** We implement CGD and conduct experiments on four benchmark datasets. The results demonstrate that CGD consistently achieves superior privacy, accuracy and fairness performance in all settings. The performance also remains stable even with a relatively large number of participants (*e.g.,* 112,000 participants on the MNIST dataset and 32,000 participants on the CIFAR-10 dataset).

**Open Science.** To the best of our knowledge, the paper presents the first practical privacy-preserving and fairness-aware federated learning against malicious clients for critical infrastructure protection and resilience. We release the source code of CGD at https://github.com/CGD-release/cgd to foster further research.

## 2 BACKGROUND AND RELATED WORK

In this section, we introduce the background and recent works that are related to our study.

### 2.1 Privacy-preserving Federated Learning

In FL, each local participant $l \in \mathcal{L}$ holds a subset of the training samples, denoted by $\xi_l$. Taking FedSGD [33, 34, 67] as an example, at each iteration, a random subset $\xi_{l,k} \subseteq \xi_l$ from a random participant $l$ is selected. The participant $l$ then computes the gradient with respect to $\xi_{l,k}$, which can be written as $\nabla_k^l$, and shares the gradient

with the aggregator (server). All participants (or the aggregator) can thus take a gradient descent step by

$$w_{k+1} \leftarrow w_k - \alpha_k \frac{1}{|\xi_{l,k}|} \nabla_k^l. \tag{1}$$

The existing privacy-preserving methods for FL mainly fall into two broad categories, *i.e., secure aggregation* and *differential privacy*.
**Secure Aggregation.** Secure aggregation typically employs cryptographic mechanisms such as homomorphic encryption (HE) [13, 44, 57] and/or secure multiparty computation (MPC) [8, 21, 22, 43, 48, 69] to securely work out the gradient $\nabla F(w_k, \xi_{l,k})$ without revealing local gradients. Each participant obtains the $\nabla F(w_k, \xi_{l,k})$ only, rather than the local gradient of any other participant, and then all participants take a gradient descent step by Equation 1.

Existing FL frameworks employing HE/MPC are mainly based on federated SGD [8, 69]. All their participants share and update one plain global model, which may incur high overhead. In CGD, we utilize MPC to securely calculate the sum of local updates, each of which is computed based on a private and different confined model. This ensures the secrecy of individual updates while devolving the computational complexity to the local gradients.
**Learning with Differential Privacy (DP).** One of the strongest privacy standards that can protect FL models from a variety of privacy attacks is differential privacy [2, 16, 23, 27, 29, 58, 61, 70, 71]. Here we employ the notion of $(\epsilon, \delta)$-differential privacy which is commonly used in machine learning.

DEFINITION 1. $(\epsilon, \delta)$-**differential privacy.** *A randomized mechanism $\mathcal{M} : \mathcal{D} \rightarrow \mathcal{R}$ with domain $\mathcal{D}$ and range $\mathcal{R}$ satisfies $(\epsilon, \delta)$-differential privacy if for any two adjacent inputs $d, d' \in \mathcal{D}$ and for any subset of outputs $S \subseteq \mathcal{R}$ it holds that*

$$Pr[\mathcal{M}(d) \in S] \le e^{\epsilon} Pr[\mathcal{M}(d') \in S] + \delta. \tag{2}$$

The definition permits the possibility that plain $\epsilon$-differential privacy can be broken with a preferably subpolynomially small probability $\delta$ when applying the Gaussian noise mechanism [2, 16]. The common practice of achieving differential privacy in FL is to add noise calibrated to the local updates, *i.e.,* $\nabla F$'s sensitivity $\mathcal{S}_{\nabla F}^2$. As such, a differentially private learning framework can be achieved by updating parameters with perturbed local gradients at each iteration, for example, to update parameters as

$$w_{k+1} \leftarrow w_k - \alpha_k \frac{1}{|\xi_{l,k}|} (\nabla F(w_k, \xi_{l,k}) + \mathcal{N}(0, \mathcal{S}_{\nabla F}^2 \cdot \sigma^2)), \tag{3}$$

where $\mathcal{N}(0, \mathcal{S}_{\nabla F}^2 \cdot \sigma^2)$ is the Gaussian distribution with mean 0 and standard deviation $\mathcal{S}_{\nabla F} \cdot \sigma$.

Recent studies introduced user-level differential privacy in federated learning [5, 45], which ensures an adversary cannot distinguish the presence/absence of *all* the data belonging to a particular participant. The guarantee is based on a specific notion of *user-adjacent datasets, i.e.,* two datasets are user-adjacent if one can be formed by adding or removing *all* of the examples associated with a single user from the other [45]. In this study, the privacy analysis framework follows the traditional notion of differential privacy with respect to a single example, incorporating a more generic learning scenario.

## 2.2 Fairness in FL

In the FL scenario, many factors can affect the fairness of the client, *e.g.,* the accuracy performance in the presence of high data heterogeneity. For example, one common approach to mitigating the straggler problem (FL training progress moves at the pace of the slowest clients) in synchronous federated learning is to begin training on a larger number of clients than necessary and subsequently remove the slowest clients from the round [7]. However, this may inherently cause a fairness problem, as the data from the straggler clients is not fully involved in the training phase. Recent research [56] looks into network-level attacks against FL models, such as packet drop attacks, where an attacker controls over the network traffic of multiple clients and further impacts the accuracy of the global model. We argue that such a network-level availability attack also poses a threat to the fairness of federated learning models by preventing data communication between victim clients and the model.

To mitigate this, personalized FL is an emerging line of research that aims to improve fairness in federated learning. The paradigm allows participants to adapt the local models to fit their specific data distribution. Commonly used approaches of personalized FL include meta-learning (*i.e.,* training a global model while fine-tuning, regularizing, and/or interpolating the local models) [17, 39, 41] and partially local methods (*i.e.,* partitioning the model to a shared backbone component and the adaptive personal ones) [15, 53, 60]. Recently, Li *et al.* [39] propose Ditto, which is a scalable federated multi-task learning framework towards fairness in FL. It encourages the personalized models to be close to the optimal global model through local training with regularization. Following this work, we provide a formal definition of *fairness* in FL.

DEFINITION 2. $\gamma$-**Fairness.** *A federated learning system satisfies $\gamma$-fairness, if $std\{F_l(w)\}_{l \in \mathcal{L}} \le \gamma$, where $F_l(\cdot)$ denotes the test loss on participant $l \in \mathcal{L}$, and $std\{\cdot\}$ denotes the standard deviation.*

Our work fundamentally differs from these approaches by eliminating the shared global model in FL. Our strategy instead allows the proprietary confined models to further adapt to the heterogeneity in locality and push up the peer-to-peer communication layer away from locality detailed as bounded asynchrony in § 5.1, which provides inherent benefits of fairness.

## 3 CONFINED GRADIENT DESCENT

In this section, we formalize this problem and present CGD's architecture, threat model, and the optimization algorithm.

## 3.1 Problem Formulation

Consider a centralized dataset $\xi = \{(x_i, y_i)\}_{i=1}^n$ consisting of $n$ training samples. The goal of machine learning is to find a model parameter $w$ such that the overall loss, which is measured by the distance between the model prediction $h(w, x_i)$ and the label $y_i$ for each $(x_i, y_i) \in \xi$, is minimized. This is reduced to solving the following optimization problem: $\arg \min_w \frac{1}{n} \sum_{i=1}^n F(w; \xi) + \beta z(w)$, where $F(w; \xi)$ stands for the loss function, and $z(w)$ is the regularizer.

In the context of FL, we have a system of $m$ local participants, each of which holds a private dataset $\xi_l \subseteq \xi$ ($l \in [1, m]$) consisting of a part of the training dataset. The part could be a part of training

samples (*i.e.,* the *horizontal partition*), a part of features that have common entries (*i.e.,* the *vertical partition*), or both.

**Optimization Objective of CGD**. CGD starts with a set of discrete points $\{w_1^l\}$. Assume the training takes $T$ iterations, and let $w_k^l$ denote the confined model of participant $l$ at the $k^{th}$ iteration, where $k \in [1, T]$. Let

$$g^l(w_k^l, \xi_l) = \frac{1}{|\xi_l|} \nabla F(w_k^l, \xi_l) \tag{4}$$

represent the local gradient with respect to $\xi_l$. The objective of CGD is to learn a separate confined model for each participant, by synthesizing the local gradients in a secure way. When CGD converges, participant $l$ derives its own final confined model $w_*^l$. We use $w_*$ to denote the optimal solution of centralized training (*i.e.,* the *centralized model*). CGD aims to make $w_*^l$ located at a neighborhood of $w_*$ within a bounded gap.

## 3.2 CGD Architecture

As shown in Figure 3(a), in traditional FL, every participant holds its local training dataset, and all of them update the *same* global model $w_k$ via a parameter server using their local models/gradients. The local gradients $\nabla F(w_k, \xi_{l,k})$ can be protected via either secure aggregation [8, 43, 44, 48, 57, 69] or differential privacy mechanisms [2, 16, 29, 58, 61, 70].

As seen in CGD (Figure 3(b)), each participant learns its own confined model (represented by different colors), *i.e.,* each $w_k^l$ is *different* and *private*. The model updating in CGD synthesizes the aggregated information by summing up the local gradients from a subset of participants. To better position CGD, we summarize and compare existing privacy-preserving federated learning approaches with our CGD in Table 1.

**Desired properties.** CGD should have the following properties:

- Advantageous privacy-utility tradeoff. CGD should not degrade the utility (*e.g.,* accuracy) while preserving the privacy of participants. Concretely, CGD pursues an advantageous tradeoff between privacy and model utility via *the secrecy of the confined models* (as detailed in Section 4).
- Robustness against membership inference attacks. Considering the emerging privacy threats introduced by membership inference attacks (MIAs), as a novel and practical federated learning approach, CGD guarantees its robustness against MIAs, including passive MIA and active MIA, through *the participant-side detect-then-drop strategy* (as detailed in Section 4).
- Fairness-awareness. CGD should provide a practical approach to achieving a uniform utility distribution, minimizing the standard deviation of test loss across participants. CGD guarantees that each confined model, when CGD converges, is at the neighborhood of the centralized model. This property retains desirable model accuracy and fairness according to Definition 2 (as proved in Section 5).
- Asynchronous federated learning enabled. In traditional synchronous FL, progress moves at the synchronous pace. In CGD, we enable a mechanism to allow bounded asynchronous optimization, *i.e.,* slow participants can upload data in later rounds, and such asynchrony is up to a bounded delay for the fairness guarantee.

## 3.3 Threat Model of Privacy and Fairness Attacks

**Threat model of privacy attacks.** We consider a membership inference adversary who aims to infer whether a given data record was used to train the model. We include two representative modes of privacy attacks in CGD, *i.e.,* passive mode and active mode. In the passive mode, the adversary makes observations of the genuine computations during the federated learning; while in the active mode, the adversary can further actively modify its local updates with respect to the victim samples for the inference attack [46, 50, 68]. In both modes, the adversary has white-box access to the local model, following the existing privacy-preserving federated learning frameworks [2, 46, 50, 58]. The attack participants can collude with each other, but we assume that there are at most $m - 2$ out of $m$ participants controlled by the attacker.

**Threat model of fairness attacks.** Regarding model fairness, we focus on network-level packet drop threats. They can arise from various methods, which have been studied in the context of network-level adversaries who carry out physical-layer attacks or transport-layer attacks [4, 31]. Adapted from Severi *et al.* [56], in the network-level packet drop threat, the attacker aims to decrease the accuracy of the FL model on a particular group of clients.

In this research, we particularly focus on the privacy and fairness attacks. The general poisoning attacks that aim to corrupt the overall model prediction are beyond the scope of this study.

## 3.4 CGD Algorithm

Algorithm 1 outlines the optimization process, which consists of the following steps.

- **Initialization** (Line 3). Each participant $l$ randomizes its own $w_1^l$ with an initialization function $init()$ which takes three arguments: $w_{anchor}$ denotes an anchor matrix that is shared among participants (*e.g.,* a zero matrix); $Rand^l$ denotes the randomization distribution (a typical one is a Gaussian distribution); and $\lambda^l$ is to control the distance of the initial points to the anchor, and CGD allows each participant to choose their own $\lambda^l$. On the other hand, $init()$ also represents the tradeoff of privacy and accuracy. Intuitively, higher degree of randomness leads to differing confined models, but may undermine the accuracy performance of the confined models. In the following, we define a procedure called *balanced initialization* which balance out the randomness.

  DEFINITION 3. *(Balanced initialization). Given an initialization scheme Rand, each CGD participant uses an initialization parameter $\lambda^l$ to initialize its confined model $w^l$ in the way of*

$$w_1^l = w_{anchor} + \lambda^l \cdot Rand^l \quad for\ all\ l \in (1, m). \tag{5}$$

  The key idea of the balanced initialization is to let each participant determines its own $Rand^l$ and $\lambda^l$ independently to avoid leaking the average distance among the confined models.

- **Step 1: Compute local gradients** (Line 6). Every participant computes the local gradient $g_k^l$ with respect to its current confined model $w_k^l$ and local dataset $\xi_l$ using Equation 4. We highlight that in traditional FL, each participant's local gradient is

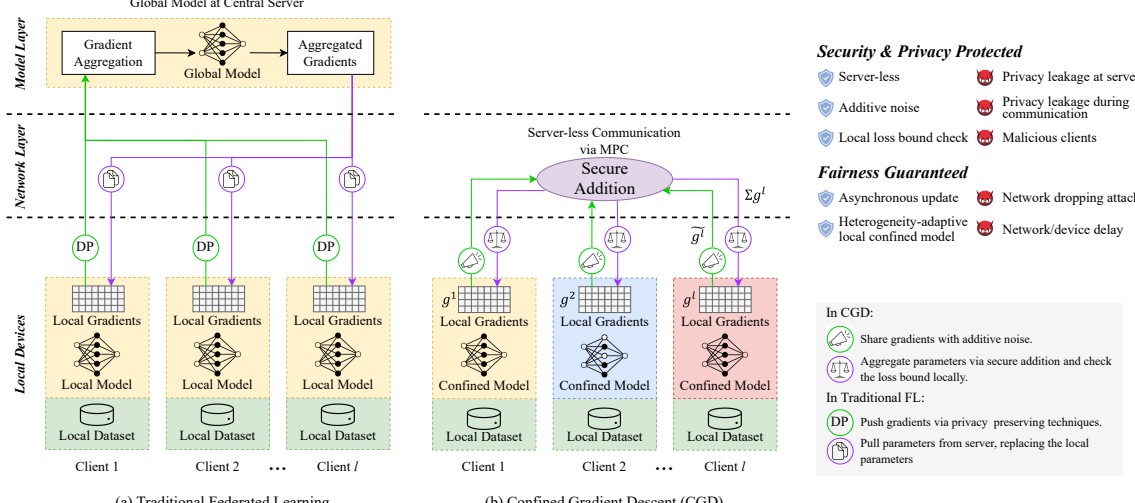

**Figure 3: Architectural comparison of CGD and the traditional federated learning.**

---

**Algorithm 1** Confined Gradient Descent Optimization

**Input:** A set of participants $l \in L$, the local training data $\xi_l$, fraction of selected participants $C$, the number of training iterations $T$, and the loss threshold $LRR$.
**Output:** Confined model for each participant $w_*^l$.

1: $k \leftarrow 1$
2: **for** each $l$ **do**
3:     $w_1^l \leftarrow init(w_{anchor}, Rand^l, \lambda^l)$
4: **while** $k \leq T$ **do**
5:     **for** participants $l \in L$ **do**
6:        $g_k^l \leftarrow F(w_k^l, \xi_l)$
7:        $\widetilde{g_k^l} \leftarrow addNoise(g_k^l, \lambda^l)$
8:        $S \leftarrow randomSelect(C, L)$
9:        $\widetilde{g_k} \leftarrow \sum_{s \in S} \widetilde{g_k^s}$
10:      $\alpha_k = chooseStepSize()$
11:      $w \leftarrow w_k^l - \alpha_k \widetilde{g_k}$
12:      $LRR_k \leftarrow F(w, \xi_l) - g_k^l$
13:      **if** $LRR_k \leq LRR$ **then**
14:        $w_{k+1}^l \leftarrow w$
15:      **else**
16:        $w_{k+1}^l \leftarrow w_k^l$
17:    $k \leftarrow k + 1$

---

computed based on the same global model at the current iteration. In contrast, the local gradients in CGD are computed from private confined models which are different among participants.

- **Step 2: Add noise** (Line 7). In this step, CGD allows each participant to add noise to its local gradients with an $addNoise(g_k^l, \lambda^l)$ function. The perturbed local gradient is denoted as $\widetilde{g_k^l}$. This step aims to facilitate the comparison of privacy preservation between CGD and the traditional DP mechanism. We show that less noise is required for the same level of privacy guarantee, resulting in better privacy-accuracy tradeoff (detailed in Section 4).

- **Step 3: Secure addition of gradients** (Line 9). A subset of participants $S$ (randomly selected with the fraction parameter $C$) jointly compute the aggregated gradient $\widetilde{g_k}$, which is later used for loss bound checking. In order to preserve the secrecy of the confined models, we employ the additive secret sharing [6] for securely evaluating the sum of a set of secret values to avoid

releasing the addends in plain text. In Appendix G, we prove that the secrecy of the confined models is guaranteed.

- **Step 4: Confined model pre-step** (Lines 10 to 11). Every participant in the current iteration computes its own $w$ with learning rate $\alpha_k$ as a pre-step, which will be applied to the update of the confined model (or dropped) after the local loss bound check.

- **Step 5: Local loss bound checking** (Lines 12 to 16). Every participant compute loss $LRR_k$ in this round if $w$ is applied, *i.e.,* the change of model's prediction performance in terms of the loss function value, and takes a descent step on $w$ if the performance degradation is less than the local loss threshold $LRR$.

### 3.5 Implementation and Benchmarks

**Datasets.** We implement and evaluate CGD's performance on four benchmark datasets, *i.e.,*, MNIST [36], CIFAR-10 [35], Location30 [66] and Hourly Energy Consumption (HEC) [49]. We give the details of the datasets and models in Appendix A.

**Benchmarks.** We compare CGD's performance in terms of privacy and fairness with the benchmarks in the following.

*i) Differentially private benchmarks.* We compare CGD with FL with DP mechanisms, *i.e.,* local updates perturbation. Specifically, we take the following two state-of-the-art differentially private FL schemes in our experiments. The settings of all benchmarks are listed below.

- **SDP** [2]. The standard Gaussian mechanism is based on the moment accountant proposed by Abadi *et al.* [2].
- **BDP** [63]. We also compare the state-of-the-art Bayesian Differential Privacy (BDP) which takes into account that training samples in ML are typically drawn from the same distribution, and it thus calibrates noise to the data distribution and provides tight privacy guarantees.

We further compare the performance of CGD with **Centralized training** (*i.e.,* training on aggregated data), **Local training** (*i.e.,* training on each participant's local data only), and **FedAvg** [34].
*ii) Preserving privacy against poisoning MIA in the active mode.* We implement the state-of-the-art poisoning membership

inference attack against FL [46, 50, 68], in which the adversary pushes updates towards ascending the gradients of the global model. ***iii) Fairness benchmark.*** For fairness benchmarking, we compare with Ditto [39], which is the state-of-the-art towards fairness in FL. It is a personalization add-on that encourages the personalized models to be close to the optimal global model through local training with regularization.

**Default FL setting.** CGD is trained using Algorithm 1 on the dataset which is partitioned and distributed to $m$ participants, where $m = (m^h \times m^v)$. By default, the initialization parameter $\lambda$ is set to 0.1, and $m$ is set to $(4 \times 4)$ in our experiments of privacy and fairness analysis.

## 4  PRIVACY ANALYSIS

In this section, we first analyze CGD's privacy preservation within the framework of DP in Sections 4.1. We then conduct experimental evaluation to demonstrate privacy-utility performance of CGD as well as its capacity in defending MIA (Section 4.2).

### 4.1  Privacy Bound in CGD

In our analysis, we let the privacy cost be:

$$\mathcal{P} = \log\left(\frac{Pr(\mathcal{M}(d) = S)}{Pr(\mathcal{M}(d') = S)}\right)$$

.

We use the Chernoff bound, which is widely adopted in various privacy accounting techniques [2, 47, 63], to convert the Inequality 2 as

$$Pr(\mathcal{P} \geq \epsilon) \leq \frac{\mathbb{E}[e^{c\mathcal{P}}]}{e^{c\epsilon}}. \tag{6}$$

The inequality holds for any $c$, and it determines the tightness of the bound (*i.e.*, minimizes $\delta$ and/or $\epsilon$). We then define

$$\mathcal{L}(\mathcal{M})_c = \mathbb{E}\left[e^{c\mathcal{P}}\right] = \mathbb{E}\left[\left(\frac{Pr(\mathcal{M}(d) = S)}{Pr(\mathcal{M}(d') = S)}\right)^c\right]. \tag{7}$$

Inspired by the moments accountant [2], we aim to bound $\mathcal{L}(\mathcal{M})_c$, and we can then use the Chernoff inequality to transform the bound to $(\epsilon, \delta)$-differential privacy. To bound $\mathcal{L}(\mathcal{M})_c$, we first analyze the relationship of the randomness in CGD and the traditional noise in DP framework based on Gaussian mechanism [16].

Existing DP framework adds noise to the local gradients which are computed from the same global model. The noise, which we refer to as the *standard noise*, denoted as $\sigma_{standard}$ is required to be added in each single training iteration [2, 16]. In contrast, the randomness in the confined models, which is protected by additive secret sharing scheme, is incorporated to the local gradient's calculation (In Appendix G, we provide a security analysis to demonstrate that the confidentiality of the confined models is guaranteed against an honest-but-curious white-box adversary who may control $t$ out of $m$ participants, where $t \leq m - 2$). This fact can be utilized to save the amount of additive noise in each iteration to reach the same $\epsilon$. Intuitively, such randomness can be regarded as inherent noise which allows the standard additive noise $\sigma_{standard}$ to be reduced.

In CGD, each participant conducts computation based on different models with the randomness of $\mathcal{N}(0, \lambda^l)$. Thus, each $w^l$ satisfies $(\epsilon', \delta)$-differential privacy, where $\epsilon' \geq \sqrt{2 \ln \frac{1.25}{\delta \lambda^l}}$ by standard arguments (Theorem 3.22 in [16]). We then have $g^l$ satisfying $(\epsilon', \delta)$-differential privacy based on the post-processing property of

**Table 2: Comparison of privacy bounds between CGD and DP-FL.**

| | MNIST | | CIFAR-10 | | Location30 | | Energy HEC | |
|---|---|---|---|---|---|---|---|---|
| | Test Acc | $(\epsilon,\delta)$-DP | Test Acc | $(\epsilon,\delta)$-DP | Test Acc | $(\epsilon,\delta)$-DP | MSE | $(\epsilon,\delta)$-DP |
| **CGD** | 95.00% | $(\mathbf{0.41}, 10^{-5})$ | 73.00% | $(\mathbf{0.35}, 10^{-5})$ | 75.80% | $(\mathbf{0.75}, 10^{-5})$ | 0.0062 | $(\mathbf{0.80}, 10^{-5})$ |
| **BDP [63]** | 95.00% | $(0.95, 10^{-5})$ | 73.00% | $(0.76, 10^{-5})$ | 56.04% | $(329.12, 10^{-5})$ | 0.0082 | $(1.46, 10^{-5})$ |
| **SDP [2]** | 95.00% | $(2.00, 10^{-5})$ | 70.00% | $(4.00, 10^{-5})$ | 56.04% | $(329.29, 10^{-5})$ | 0.0082 | $(1.46, 10^{-5})$ |

DP. We thus can construct $g^l$ (clipped) as the sum of a $g^l_{anchor}$ and $\mathcal{N}(0, \lambda^l)$, and rewrite Line 7 in Algorithm 1 as

$$\begin{aligned}
\widetilde{g^l} &= g^l/\max(1, \|g_1^l\|_2) + \mathcal{N}(0, \sigma^2) \\
&= g^l_{anchor} + \mathcal{N}(0, \lambda^l) + \mathcal{N}(0, \sigma^2) = g^l_{anchor} + \mathcal{N}(0, \lambda^l + \sigma^2)
\end{aligned} \tag{8}$$

Therefore, we have

$$\begin{aligned}
\widetilde{g} &= \sum_{l=1}^{m} \widetilde{g^l} = \sum_{l=1}^{m} g^l_{anchor} + \sum_{l=1}^{m} \mathcal{N}(0, \lambda^l + \sigma^2) \\
&= \sum_{l=1}^{m} g^l_{anchor} + \mathcal{N}(0, \sum_{l=1}^{m} \lambda^l + m\sigma^2)
\end{aligned} \tag{9}$$

With this, we give the following lemma.

LEMMA 1. *For any positive integer $c \geq 1$, CGD with an addNoise function defined as $\widetilde{g_k^l} = g_k^l/\max(1, \|g_k^l\|_2) + \mathcal{N}(0, \sigma^2)$ satisfies*

$$\log \mathcal{L}(\mathcal{M})_c \leq \frac{c(c+1)}{\sum_{l=1}^{m} \lambda^l + m\sigma^2} + O\left(\frac{c^3}{(\sum_{l=1}^{m} \lambda^l + m\sigma^2)^{3/2}}\right) \tag{10}$$

We give the proof in Appendix B.

THEOREM 1. *Given the number of training iteration $T$, there exists a constant $a$ such that CGD is $(\epsilon, \delta)$-differentially private for any $\delta > 0$ if we choose*

$$\sigma \geq 2\sqrt{\frac{Ta \log \frac{1}{\delta}}{m\epsilon^2} - \mathbb{E}[\lambda^l]} \tag{11}$$

We give the proof in Appendix C.

We remark that the result of Theorem 1 does not take into consideration the privacy amplification via sub-sampling because the selection of the training samples at each iteration can be non-IID in the federated learning. As such, the result can be regarded as the worst case in which the batch gradient descent is conducted. In Appendix D, we also provide a tighter bound of CGD's privacy preservation with IID data.

### 4.2  Empirical Results on Privacy

**Privacy Utility *vs.* Accuracy Performance**. Table 2 summarizes the privacy bounds in CGD, BDP [63] and SDP [2] respectively, when they reach the same accuracy level. CGD retains a lower $\epsilon$ when reaching the same accuracy level as others. It achieves $(0.41, 10^{-5})$-DP on MNIST, $(0.35, 10^{-5})$ on CIFAR-10, $(0.75, 10^{-5})$ on Location30, and $(0.80, 10^{-5})$ on Energy HEC, all stronger than BDP and SDP.

Figure 4 illustrates the performance of CGD on the test accuracy compared to the FedAvg and local training. For MNIST, with the privacy bounds at $(0.54, 10^{-5})$, CGD achieves 96.9% in test accuracy, which is close to 97.54% of FedAvg and outperforms of 71.77% of the local training. For CIFAR-10, FedAvg reaches the test accuracy of 75.75%, which is taken as our baseline.[2] In line with the performance on MNIST, with the privacy bounds at $(0.36, 10^{-5})$,

---

[2]We note that by making the network deeper or using other advanced techniques, better accuracy can be obtained, given the state-of-the-art being 99.37% [32].

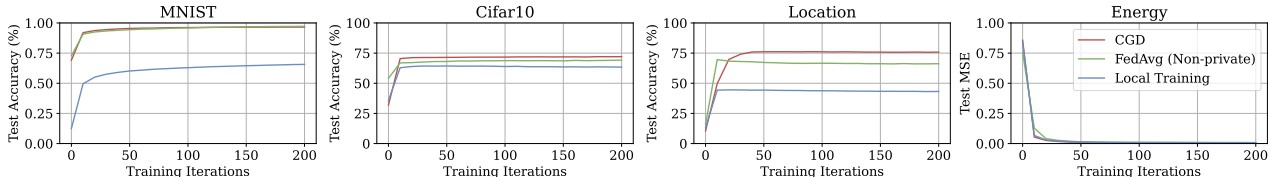

**Figure 4: CGD's test accuracy compared to FedAvg and local training.**

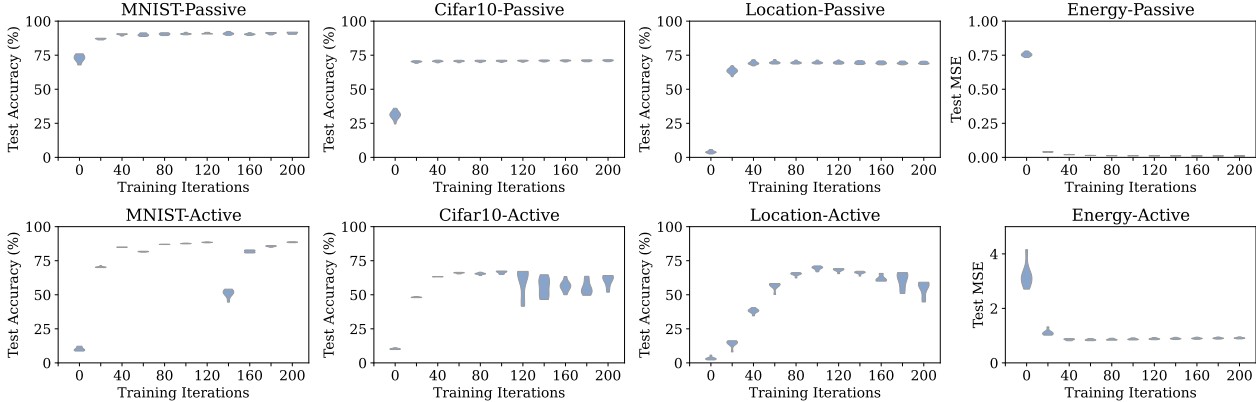

**Figure 5: The variation of test accuracy among participants in CGD.**

**Table 3: CGD's performance against active MIA.**

|  |  | MNIST | CIFAR | Location30 | Energy HEC |
|---|---|---|---|---|---|
| **Active MIA** | CGD | **51.30%** | **50.00%** | **55.10%** | **50.10%** |
| **Success Rate** | FedAvg | 54.30% | 60.30% | 88.30% | 72.87% |
| **Model Performance** | CGD | **92.00%** | **66.90%** | **68.82%** | **0.94** |
|  | FedAvg | 86.80% | 45.60% | 33.33% | 1.14 |

**Table 4: CGD's performance of fairness (Definition 2): Standard deviation of test accuracy on MNIST, CIFAR, Location30, and standard deviation of MSE on Energy HEC among participants.**

|  |  | MNIST | CIFAR | Location30 | Energy HEC |
|---|---|---|---|---|---|
| **Clean** | CGD | **0.0013** | **0.0043** | **0.0023** | **0.0001** |
| **(honest-but-curious)** | Ditto | 0.0082 | 0.1223 | 0.0698 | 0.0051 |
| **Active MIA** | CGD | **0.0043** | **0.1370** | **0.0523** | **0.0005** |
|  | Ditto | 0.1974 | 0.1720 | 0.1657 | 0.0103 |

CGD achieves 74.30% in test accuracy, while the local training only reaches 53.67%. For Location30 and Energy HEC, CGD achieves 75.8% of test accuracy with $(0.75, 10^{-5})$-DP and 0.0062% of MSE with $(0.80, 10^{-5})$, respectively, outperforming the test accuracy of 73.27%, 0.0072 in FedAvg, and 42.80%, 0.007 in the local training.

**Robustness against Active MIA**. We set $m = (8 \times 4)$, *i.e.*, 32 participants in the experiments and 20% of them are controlled by the poisoning adversary, following the standard setting in the literature [11, 18, 30]. Table 3 summarizes CGD's performance against the active membership inference attack. In general, CGD significantly mitigates the attack, reducing the attack success rate to around 50% (*i.e.*, random guess of member and non-member) across all datasets. It also retains the model prediction performance: it achieves 92.00%, 66.90%, 68.82% of test accuracy, and 0.94 of MSE on MNIST, CIFAR-10, Location30, and Energy HEC, respectively,

whereas the test accuracy in FedAvg (without the protection of CGD) reduces significantly due to the poisoning.

## 5 FAIRNESS ANALYSIS

In this section, we first conduct a formal analysis on the bound of the distance between an arbitrary confined model $w_*^l$ learned by CGD and the centralized model $w_*$, demonstrating the fairness-convergence guarantee (Sections 5.1- 5.2). We then conduct experiments to demonstrate CGD's fairness on real-world datasets, comparing with the state-of-the-art fair FL method [39] (Section 5.3).

### 5.1 Bounded Asynchrony in CGD

In CGD, a participant does not take the descent step when detecting any suspicious updates against it. This results in the asynchronous optimization, indicating that any two participants may be at different iteration rounds. For any participant $l$ at iteration $k$, the confined model $w_k^l$ is updated as

$$w_{k+1}^l = w_k^l - \alpha_k \sum_{j=1}^{S_t} \widetilde{g}^j(w_{k-\tau_k^j}^j, \xi_j), \quad (12)$$

where $\tau_k^j$ denotes the *staleness*, *i.e.,* how many iterations of a delay between the participants $j$ and $l$ at $k^{th}$ iteration. We show that, when $C \le 1 - P_{\mathcal{A}}$ (recall $C$ is a fraction parameter of $m$ participants, $\mathcal{A}$ denotes the set of malicious participants, and $P_{\mathcal{A}}$ denotes the malicious proportion, *i.e.,* $P_{\mathcal{A}} = |\mathcal{A}|/m$), the staleness in CGD is bounded (*i.e.,* $\tau_k^j \le \tau$), with a preferably subpolynomially small probability being broken (see Appendix H).

Thus, the bounded staleness of $\tau$ can also be utilized to make CGD resistant to network-level attacks against FL models, such as a packet drop attack, *i.e.,* CGD is tolerant to the drop up to $\tau$.

## 5.2 Fairness-Convergence Theorem

We now give an upper bound of the distance between any participant's $w_k^l$ and $w_*$, demonstrating the convergence rate and the fairness guarantee in CGD. Our analysis investigates a *regret function* $\mathcal{R}$, which represents the difference between the CGD's loss and the loss of the centralized model. $\mathcal{R}$ is defined as

$$\mathcal{R} = \frac{1}{T} \sum_{k=1}^{T} (F(w_k^l) - F(w_*)). \tag{13}$$

Using it, we present our Fairness-Convergence Theorem below, establishing the guarantee of CGD converging to the neighborhood of the global optimum, where the loss against the optimum is bounded.

We first present the assumption on a bounded solution space which is widely used in related studies [10, 29].

ASSUMPTION 1. *(Bounded solution space). The set of $\{w_k^l\}|_{k \in [1,T]}$ is contained in an open set over which $F$ is bounded below a scalar $F_{inf}$, such that (i) there exists a $D > 0$, s.t., $\|w_k^l - w_*\|_2^2 \le D^2$ for all $k$, and , and (ii) there exists a $G > 0$, s.t., $\|g^l(w_k^l, \xi_l)\|_2^2 \le G^2$ for all $w_k^l \subset \mathbb{R}^d$.*

We also follow prior works [29, 62, 65] in which Lipschitz continuity and convexity are applied. Such assumptions are based on the observation that a trainable deep neural network, *e.g.*, ResNets-110 (containing 110 layers) with shortcut/skip connections, requires flat minima and wide regions of apparent convexity, while chaotic loss surfaces, with dramatic non-convexities where the gradients are uninformative, are hard to train [38]. For a detailed discussion on Lipschitz continuity used in the following theorem, we refer readers to Chaudhuri et al. [12].

THEOREM 2. *For any cost function that is L-lipschitz and satisfies convexity and Assumption 1, assuming that the learning rate meets $\alpha_k = \frac{\alpha}{(k+\mu T)^2}$ ($0 < \mu < 1$), and a bounded staleness of $\tau$, the CGD optimization gives the regret $\mathcal{R}$*

$$\mathcal{R} = O(\gamma + \frac{1}{\mu T + 1} + \frac{\ln|\mu T + 1|}{T} + \frac{1}{T(1 - \tau + T)}), \tag{14}$$

*where $\gamma = m\| \underset{j \in m}{\mathbb{E}} [w_1^l - w_1^j] \|$.*

Below we discuss the implications of the theorem, and leave its proof to Appendix E.

**Implication #1: Convergence rate**. $\frac{1}{\mu T+1}$, $\frac{\ln|\mu T+1|}{T}$ and $\frac{\ln|\mu T+1|}{T} + \frac{1}{T(1-\tau+T)}$ approach 0 as $T$ increases, implying that CGD will converge toward the optimum. The convergence rate is affected by the staleness bound $\tau$, *i.e.*, a larger $\tau$ will lead to a slower rate. It can also be adjusted by the parameter $\mu$, named *diminishment factor*.

**Implication #2: Bounded optimality gap**. When CGD converges, the gap between the confined models and the centralized model is bounded by $\gamma$, implying the fairness guarantee. The gap is affected by the variance of $w_1^j$. We show that, CGD keeps robust (in terms of test accuracy) even when the participants' $\lambda^l$s differ by two orders of magnitude.

## 5.3 Empirical Results on Fairness

Figure 5 shows the fairness performance of CGD during the training iterations. When CGD converges, the standard deviation (SD) of

test accuracy across participants achieves 0.0013, 0.0043, 0.0023, and 0.0001 on MNIST, CIFAR-10, Location30, and Energy HEC in the honest-but-curious scenario, and 0.0043, 0.1370, 0.0523, and 0.0005 on these datasets against active membership inference.

In Table 4, we compare the performance of CGD with Ditto [39]. In the passive scenario (which is in line with the "clean data" setting in Ditto), CGD reduces the standard deviation of test accuracy across participants by 0.0069, 0.1180, 0.0675, and 0.0050 (MSE) on MNIST, CIFAR-10, Location30, and Energy HEC, respectively. In the active scenario (which is in line with the "data poisoning attack" setting in Ditto), CGD reduces the standard deviation of test accuracy by 0.1931, 0.0350, 0.1134, and 0.0098 on these datasets.

## 6 ABLATION STUDIES OF PERFORMANCE

### 6.1 Scalability

In this section, we investigate how the number of participants affects CGD's privacy and accuracy performance. CGD's performance remains relatively stable as the participant number grows, even with a large number of participants. With 112,000 participants (*i.e.*, $m = (1000 \times 112)$) for MNIST, CGD still achieves 95.78% in test accuracy, close to 97.54% of the centralized baseline. With 32,000 participants (*i.e.*, $m = (1000 \times 32)$) for CIFAR, CGD achieves 67.67% in test accuracy, which is also close to the centralized baseline (75.72%). We give more experimental results in Appendix I.1.

### 6.2 Effect of the Initialization

We conduct experiments to investigate the effect of initialization parameter $\lambda$. We let each participant randomly select its own $\lambda$ based on the uniform distribution within the range from 0.001 to 0.1. The performance of CGD is still comparable with the centralized mode. For example, with $m = (1000 \times 112)$ participants on MNIST and $m = (1000 \times 32)$ on CIFAR, CGD achieves 97.07% and 71.01% in the test accuracy respectively. This suggests that CGD keeps robust when the $\lambda$s of its participants differ by two orders of magnitude. This provides sufficient randomness in the balanced initialization strategy. More results of the investigate on the parameter $\lambda$ is given in Appendix I.2.

In Appendix I.3, we also provide additional results of the effect of the diminishment factor $\mu$.

## 7 CONCLUSION

We have presented CGD, a novel optimization algorithm for learning confined models to enhance privacy and fairness for federated learning. It achieves tighter differential privacy bounds than the state-of-the-art DP-FL frameworks. CGD is also effective in defending against poisoning MIA in the active mode, reducing the attack accuracy to 50%. We also formally prove the convergence-fairness of CGD optimization. Since we focus on the privacy and fairness of malicious clients or local models, security attacks such as intentionally injecting poisoned data and uploading backdoored local models on CGD provide avenues for our future research. We hope that CGD developed in this paper could be steadily leaving its infancy for building protection and resilience for critical infrastructure.

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

# APPENDIX

# A DATASETS AND MODELS

In the following, we give the details of the datasets and models.

- **MNIST** [36]. It is a benchmark dataset for handwritten digit recognition. It has $60,000$ training samples and $10,000$ test samples, each with 784 features representing $28 \times 28$ pixels in the image. We use a convolutional neural network (CNN) with ReLU of 256 units and cross-entropy loss.

- **CIFAR-10** [35]. The dataset consists of $60,000$ $32 \times 32$ colour images in 10 classes (*e.g.*, airplane, bird, and cat). We follow the settings of Abadi *et al.* [2], which treats the CIFAR-10 as the private dataset and CIFAR-100 as a public dataset. CIFAR-100 has the same image types as CIFAR-10, and it has 100 classes containing 600 images each. We use CIFAR-100 to train a network with the ResNet-56 architecture [28], and freeze the parameters of the convolutional layers and retrain only the last FC layer on CIFAR-10.

- **Location30** [66]. This dataset contains mobile users' "check-in" data in the Foursquare social network. In our experiments, we use the processed dataset created by Shokri *et al.* [59] which contains 30 classes and 5,010 user profiles, each with 446 binary features. The model architecture is a FCN with layer sizes $\{446, 512, 128, 30\}$.

- **Hourly Energy Consumption (HEC)** [49]. The dataset contains 121,275 historical hourly energy consumption records (in MegaWatts) of several major electricity distribution companies in the United States over 10 years. It is used for time-series forecasting tasks, *i.e.*, to use early sequences of energy consumption values and predict the future value for sequences of length $T$. The model architecture is LSTM, where we use Mean Squared Error (MSE) to measure the accuracy of regression prediction.

# B PROOF OF LEMMA 1

PROOF. Let $v_0$ denote the pdf of $\mathcal{N}(0, \sigma'^2)$ and let $v_1$ denote the pdf of $\mathcal{N}(1, \sigma'^2)$, where $\sigma'^2 = \sum_{l=1}^{m} \lambda^l + m\sigma^2$. Without loss of generality, we have:

$$\mathcal{M}(d) \sim v_0, \mathcal{M}(d') \sim v_1$$

.We aim to demonstrate the bound of

$$\max\left(\mathbb{E}_{x \sim v_1}\left[(v_1(x)/v_0(x))^c\right], \mathbb{E}_{x \sim v_0}\left[(v_0(x)/v_1(x))^c\right]\right).$$

We will prove one case and the method of proving the other case is similar.

$$
\begin{aligned}
\mathbb{E}_{x \sim v_1}&\left[(v_1(x)/v_0(x))^c\right] = \mathbb{E}_{x \sim v_0}\left[(v_1(x)/v_0(x))^{c+1}\right] \\
&= \mathbb{E}_{x \sim v_0}\left[\left(1 + \frac{v_1(x) - v_0(x)}{v_0(x)}\right)^{c+1}\right] \\
&= \sum_{i=0}^{c+1}\binom{c+1}{i}\mathbb{E}_{x \sim v_0}\left[\left(\frac{v_1(x) - v_0(x)}{v_0(x)}\right)^i\right] \\
&= \binom{c+1}{0}\mathbb{E}_{x \sim v_0}\left[\left(\frac{v_1(x) - v_0(x)}{v_0(x)}\right)^0\right] \\
&\quad + \binom{c+1}{1}\mathbb{E}_{x \sim v_0}\left[\left(\frac{v_1(x) - v_0(x)}{v_0(x)}\right)^1\right] \\
&\quad + \binom{c+1}{2}\mathbb{E}_{x \sim v_0}\left[\left(\frac{v_1(x) - v_0(x)}{v_0(x)}\right)^2\right] + \dots
\end{aligned}
\tag{15}
$$

The first term of the binomial expansion is 1, and the second term is

$$\mathbb{E}_{x \sim v_1}\left[\left(\frac{v_0(x) - v_1(x)}{v_1(x)}\right)^1\right] = \int_{-\infty}^{\infty} v_1(x)\frac{v_0(x) - v_1(x)}{v_1(x)}\, dx$$

$$= \int_{-\infty}^{\infty} v_0(x) - v_1(x)\, dx = 1 - 1 = 0$$

The third term is

$$\mathbb{E}_{x \sim v_0}\left[\left(\frac{v_1(x) - v_0(x)}{v_0(x)}\right)^2\right] = \mathbb{E}_{x \sim v_0}\left[\left(\frac{v_1(x)}{v_0(x)} - 1\right)^2\right]$$

$$= \mathbb{E}_{x \sim v_0}\left[\left(\frac{\frac{1}{\sigma'\sqrt{2\pi}}e^{-\frac{1}{2}\left(\frac{x-1}{\sigma'}\right)^2}}{\frac{1}{\sigma'\sqrt{2\pi}}e^{-\frac{1}{2}\left(\frac{x}{\sigma'}\right)^2}} - 1\right)^2\right]$$

$$= \mathbb{E}_{x \sim v_0}\left[\left(\exp\left(\frac{2x-1}{2\sigma'^2}\right) - 1\right)^2\right]$$

$$= \mathbb{E}_{x \sim v_0}\left[\exp\left(\frac{4x-2}{2\sigma'^2}\right)\right] - 2\mathbb{E}_{x \sim v_0}\left[\exp\left(\frac{2x-1}{2\sigma'^2}\right)\right] + 1$$

$$= \exp\left(\frac{-2}{2\sigma'^2}\right)\int_{-\infty}^{\infty}\frac{1}{\sigma'\sqrt{2\pi}}e^{-\frac{1}{2}\left(\frac{x}{\sigma'}\right)^2}e^{\left(\frac{4x}{2\sigma'^2}\right)}\, dx$$

$$\quad - 2\exp\left(\frac{-1}{2\sigma'^2}\right)\int_{-\infty}^{\infty}\frac{1}{\sigma'\sqrt{2\pi}}e^{-\frac{1}{2}\left(\frac{x}{\sigma'}\right)^2}e^{\left(\frac{2x}{2\sigma'^2}\right)}\, dx + 1$$

$$= \exp\left(\frac{-2}{2\sigma'^2}\right)\exp\left(\frac{4}{2\sigma'^2}\right)\int_{-\infty}^{\infty}\frac{1}{\sigma'\sqrt{2\pi}}e^{-\frac{1}{2}\left(\frac{x-2}{\sigma'}\right)^2}\, dx$$

$$\quad - 2\exp\left(\frac{-1}{2\sigma'^2}\right)\exp\left(\frac{1}{2\sigma'^2}\right)\int_{-\infty}^{\infty}\frac{e^{-\frac{1}{2}\left(\frac{x-1}{\sigma'}\right)^2}}{\sigma'\sqrt{2\pi}}\, dx + 1$$

$$= \exp\left(\frac{1}{\sigma'^2}\right) - 1$$

Therefore, the sum of the first three terms is

$$1 + \binom{c+1}{2}\left(\exp\left(\frac{1}{\sigma'^2}\right) - 1\right)$$

$$< \exp\left(\frac{c(c+1)}{\sum_{l=1}^{m}\lambda^l + m\sigma^2}\right) \quad \text{(with } c \geq 1\text{)}$$

The remaining terms is dominated by the $i = 3$, which is

$$O\left(\frac{c^3}{(\sum_{l=1}^{m}\lambda^l + m\sigma^2)^{3/2}}\right).$$

We omit the derivation as it can be easily obtained by the standard calculus. □

## C  PROOF OF PRIVACY THEOREM (THEOREM 1)

PROOF. The accumulated privacy cost with $T$ training iterations is

$$\log \mathcal{L}(\mathcal{M})_c^T = T \log \mathcal{L}(\mathcal{M})_c \tag{16}$$

$$< T\frac{ac(c+1)}{\sum_{l=1}^{m}\lambda^l + m\sigma^2} \tag{17}$$

Equation 16 follows the Composability (*i.e.*, Theorem 2.1 in [2]).

To make CGD $(\epsilon, \delta)$-differentially private, according to Chernoff inequality (Theorem 2.2 in [2]), it suffices that

$$T\frac{ac(c+1)}{\sum_{l=1}^{m}\lambda^l + m\sigma^2} \leq \frac{c\epsilon}{2},$$

$$\exp\left(\frac{-c\epsilon}{2}\right) \leq \delta$$

Therefore,

$$\sum_{l=1}^{m}\lambda^l + m\sigma^2 \geq \frac{2Ta(c+1)}{\epsilon} \tag{18}$$

$$\geq 4Ta\frac{\log\frac{1}{\delta}}{\epsilon^2} \tag{19}$$

Inequation 19 is from $c \geq 2\log\frac{1}{\delta}/\epsilon$. Therefore, we have

$$\sigma \geq 2\sqrt{\frac{Ta\log\frac{1}{\delta}}{m\epsilon^2} - \mathbb{E}[\lambda^l]} \tag{20}$$

□

## D  PRIVACY BOUND WITH IID DATA

In this section, we give CGD's privacy bound with IID data.

COROLLARY 1. *Given the number of training iteration $T$, the fraction of selected participants $C$ there exists a constant $a$ so that CGD is $(\epsilon, \delta)$-differentially private for any $\delta > 0$ if we choose*

$$\sigma \geq 2\sqrt{\frac{C^2 Ta\log\frac{1}{\delta}}{m\epsilon^2} - \mathbb{E}[\lambda^l]} \tag{21}$$

We omit the proof as it can be easily obtained using similar methods in Lemma 1 and Theorem 1.

## E  PROOF OF FAIRNESS-CONVERGENCE THEOREM (THEOREM 2)

Our proof aims to identify an upper bound of $\mathcal{R}$. To this end, we consider the trend of the distance from $w_k^l$ to $w_*$, which shrinks as $k$ increases, if there exists a bound. Since $w_k^l$ is updated using $\sum_{j=1}^{S_t} \widetilde{g^j}(w_{k-\tau_k^j}^j, \xi_j)$ (*i.e.*, the descent in CGD according to Equation 12 where $\tau_k^j \leq \tau$), and $w_*$ is obtained by $\nabla F$ (*i.e.*, the descent in the centralized training), the trend of the distance therefore should be related to the deviation between these two. Exploring this leads to the following lemma which describes this relationship.

LEMMA 2. *(Optimization trajectory). Let $S_{k+1} = \frac{1}{2}\|w_{k+1}^l - w_*\|_2^2$ and $S_k = \frac{1}{2}\|w_k^l - w_*\|_2^2$. Let $\nabla F(\cdot) = \frac{1}{n}\sum_{i=1}^{n}\nabla F(\cdot, \xi_i)$. We have*

$$S_{k+1} - S_k = \frac{1}{2}\|\alpha_k \sum_{j=1}^{S_t}\widetilde{g^j}(w_{k-\tau_k^j}^j, \xi_j)\|_2^2$$

$$- \alpha_k\langle w_k^l - w_*, \nabla F(w_k^l)\rangle - \alpha_k\langle w_k^l - w_*, \sum_{j=1}^{S_t}\widetilde{g^j}(w_{k-\tau_k^j}^j, \xi_j) - \nabla F(w_k^l)\rangle.$$

PROOF.

$$S_{k+1} - S_k = \frac{1}{2}\left(\|w_{k+1}^l - w_*\|_2^2 - \|w_k^l - w_*\|_2^2\right)$$

$$= \frac{1}{2}\left(\|w_k^l - \alpha_k \sum_{j=1}^{S_t} \widetilde{g^j}(w_{k-\tau_k^j}^j, \xi_j) - w_*\|_2^2 - \|w_k^l - w_*\|_2^2\right)$$

$$= \frac{1}{2}\|\alpha_k \sum_{j=1}^{S_t} \widetilde{g^j}(w_{k-\tau_k^j}^j, \xi_j)\|_2^2 - \alpha_k \langle w_k^l - w_*, \sum_{j=1}^{S_t} \widetilde{g^j}(w_{k-\tau_k^j}^j, \xi_j)\rangle$$

$$= \frac{1}{2}\|\alpha_k \sum_{j=1}^{S_t} \widetilde{g^j}(w_{k-\tau_k^j}^j, \xi_j)\|_2^2 \tag{22}$$

$$- \alpha_k \langle w_k^l - w_*, \sum_{j=1}^{S_t} \widetilde{g^j}(w_{k-\tau_k^j}^j, \xi_j) + \nabla F(w_k^l) - \nabla F(w_k^l)\rangle$$

$$= \frac{1}{2}\|\alpha_k \sum_{j=1}^{S_t} \widetilde{g^j}(w_{k-\tau_k^j}^j, \xi_j)\|_2^2 - \alpha_k \langle w_k^l - w_*, \nabla F(w_k^l)\rangle$$

$$- \alpha_k \langle w_k^l - w_*, \sum_{j=1}^{S_t} \widetilde{g^j}(w_{k-\tau_k^j}^j, \xi_j) - \nabla F(w_k^l)\rangle$$

Dividing the above equation by $\alpha_k$, we can prove the lemma. □

**Overview of the Proof of Theorem 2.** The optimization trajectory facilitates the proof of our main convergence theorem, in which we calculate a function that is greater than $\mathcal{R}$ based on the convexity of the objective function (Inequation 23). The function can be decomposed into three terms (see Equation 24) based on Lemma 2. We name the first two terms as *overall updates*, representing the sum of all model updates throughout the entire optimization. The third term is named as *gradients margin* since it measures the gap of the gradients between CGD and the centralized training. We then explore the boundedness of each term, and taking these bounds together concludes the proof.

PROOF. By the definition of the regret function (Equation 13) and convexity, we have

$$\mathcal{R} = \frac{1}{T}\sum_{k=1}^{T}(F(w_k^l) - F(w_*)) \le \frac{1}{T}\sum_{k=1}^{T}\langle w_k^l - w_*, \nabla F(w_k^l)\rangle \tag{23}$$

Applying Lemma 2 to Inequation 23 and multiplying it by $T$, we have

$$T \cdot R \le \sum_{k=1}^{T}\left(\frac{1}{2}\alpha_k\|\sum_{j=1}^{S_t} \widetilde{g^j}(w_{k-\tau_k^j}^j, \xi_j)\|_2^2\right.$$

$$\left. - \frac{1}{\alpha_k}(S_{k+1} - S_k) - \langle w_k^l - w_*, \sum_{j=1}^{S_t} \widetilde{g^j}(w_{k-\tau_k^j}^j, \xi_j) - \nabla F(w_k^l)\rangle\right)$$

$$\le \sum_{k=1}^{T}\frac{1}{2}\alpha_k\|\sum_{j=1}^{S_t} \widetilde{g^j}(w_{k-\tau_k^j}^j, \xi_j)\|_2^2 - \sum_{k=1}^{T}\frac{1}{\alpha_k}(S_{k+1} - S_k) \tag{24}$$

$$- \sum_{k=1}^{T}\langle w_k^l - w_*, \sum_{j=1}^{S_t} \widetilde{g^j}(w_{k-\tau_k^j}^j, \xi_j) - \nabla F(w_k^l)\rangle$$

Inequation 24 can be decomposed into three terms, *i.e.*,

$\sum_{k=1}^{T}\frac{1}{2}\alpha_k\|\sum_{j=1}^{S_t} \widetilde{g^j}(w_{k-\tau_k^j}^j, \xi_j)\|_2^2$ and $-\sum_{k=1}^{T}\frac{1}{\alpha_k}(S_{k+1} - S_k)$ are *overall updates*, and $-\sum_{k=1}^{T}\langle w_k^l - w_*, \sum_{j=1}^{S_t} \widetilde{g^j}(w_{k-\tau_k^j}^j, \xi_j) - \nabla F(w_k^l)\rangle$ is *gradients margin*. Next, we explore the boundedness of each term. For

the *overall updates*, we have

$$\sum_{k=1}^{T}\frac{1}{2}\alpha_k\|\sum_{j=1}^{S_t} \widetilde{g^j}(w_{k-\tau_k^j}^j, \xi_j)\|_2^2 \le \sum_{k=1}^{T}\frac{1}{2}\frac{\alpha}{(k+\mu T)^2}C^2 m^2 G^2 \tag{25}$$

$$= \frac{\alpha C^2 m^2 G^2}{2}\sum_{k=1}^{T}\frac{1}{(k+\mu T)^2} < \alpha C^2 m^2 G^2 . \tag{26}$$

Inequation 25 is based on Assumption 11, and Inequation 26 is based on the solution to the Basel problem that $\sum_{x=1}^{\infty}\frac{1}{x^2} < 2$.

$$-\sum_{k=1}^{T}\frac{1}{\alpha_k}(S_{k+1} - S_k) = \sum_{k=1}^{T}\frac{1}{\alpha_k}(S_k - S_{k+1}) \tag{27}$$

$$= \sum_{k=1}^{T}\frac{1}{\alpha_k}\left(\frac{1}{2}\|w_k^l - w_*\|_2^2 - \frac{1}{2}\|w_{k+1}^l - w_*\|_2^2\right) \tag{28}$$

$$\le \sum_{k=1}^{T}\frac{1}{2\alpha_k}\|(w_k^l - w_*) - (w_{k+1}^l - w_*)\|_2^2 \tag{29}$$

$$= \sum_{k=1}^{T}\frac{1}{2\alpha_k}\|w_k^l - w_{k+1}^l\|_2^2 = \sum_{k=1}^{T}\frac{1}{2\alpha_k}\|\alpha_k \sum_{j=1}^{S_t} \widetilde{g^j}(w_{k-\tau_k^j}^j, \xi_j)\|_2^2 \tag{30}$$

$$= \sum_{k=1}^{T}\frac{1}{2}\alpha_k\|\sum_{j=1}^{S_t} \widetilde{g^j}(w_{k-\tau_k^j}^j, \xi_j)\|_2^2 < \alpha C^2 m^2 G^2 \tag{31}$$

Inequation 29 follows reverse triangle inequality, and Inequation 31 reuses the result of the first term (Equation 26).

As determining the bound of *gradients margin* is slightly complex, we list it as the following claim and prove it in Appendix F.

CLAIM 1. *(Gradients margin).*

$$-\sum_{k=1}^{T}\langle w_k^l - w_*, \sum_{j=1}^{S_t} \widetilde{g^j}(w_{k-\tau_k^j}^j, \xi_j) - \nabla F(w_k^l)\rangle$$

$$< DL\|T\sum_{j=1}^{S_t}(w_1^l - w_1^j)\| + 2\alpha C^2 m^2 GDL(\frac{T}{\mu T + 1} + \ln|\mu T + 1|) \tag{32}$$

$$+ DLCmG\alpha\tau(\frac{1}{1 - \tau + T})$$

Combining Inequations 24, 26, 31 and Claim 1, and dividing by $T$ we obtain

$$\mathcal{R} < \frac{2\alpha m^2 G^2}{T} + DL\|\sum_{j=1}^{m}(w_1^l - w_1^j)\|$$

$$+ 2\alpha C^2 m^2 GDL(\frac{1}{\mu T + 1} + \frac{\ln|\mu T + 1|}{T})$$

$$+ DLCmG\alpha\tau(\frac{1}{T(1 - \tau + T)})$$

$$= O(\gamma + \frac{1}{\mu T + 1} + \frac{\ln|\mu T + 1|}{T} + \frac{1}{T(1 - \tau + T)})$$

□

## F PROOF OF CLAIM 1

PROOF.

$$-\sum_{k=1}^{T}\langle w_k^l - w_*, \sum_{j=1}^{S_t} \widetilde{g^j}(w_{k-\tau_k^j}^j, \xi_j) - \nabla F(w_k^l)\rangle$$

$$= \langle w_k^l - w_*, \sum_{k=1}^{T} \left( \nabla F(w_k^l) - \sum_{j=1}^{S_t} \widetilde{g^j}(w_{k-\tau_k^j}^j, \xi_j) \right) \rangle \tag{33}$$

$$\leq \| w_k^l - w_* \| \cdot \| \sum_{k=1}^{T} \left( \nabla F(w_k^l) - \sum_{j=1}^{S_t} \widetilde{g}(w_{k-\tau_k^j}^j, \xi_j) \right) \| \tag{34}$$

$$\leq \| w_k^l - w_* \| \cdot L \| \sum_{k=1}^{T} \sum_{j=1}^{S_t} (w_k^l - w_{k-\tau_k^j}^j) \| \tag{35}$$

$$\leq DL \| \sum_{j=1}^{S_t} (w_1^l - w_{1-\tau_1^j}^j) + ... + \sum_{j=1}^{S_t} (w_T^l - w_{T-\tau_T^j}^j) \| \tag{36}$$

$$= DL \| \sum_{j=1}^{S_t} (w_1^l - w_1^j) + ... + \sum_{j=1}^{S_t} (w_T^l - w_T^j)$$
$$+ \sum_{k=1}^{T} \sum_{j=1}^{S_t} \sum_{q=k-\tau_k^j}^{q=k-1} (w_{q+1}^j - w_q^j) \| \tag{37}$$

$$= DL \| T \sum_{j=1}^{S_t} (w_1^l - w_1^j)$$
$$- \sum_{k=1}^{T-1} \sum_{j=1}^{S_t} \alpha_{(T-k)} k \left( \nabla F(w_{(T-k)}^l) - \sum_{j=1}^{S_t} g^j(w_{(T-k)}^j, \xi_j) \right)$$
$$+ \sum_{k=1}^{T} \sum_{j=1}^{S_t} \sum_{q=k-\tau_k^j}^{q=k-1} (w_{q+1}^j - w_q^j) \| \tag{38}$$

$$\leq DL \Bigg( \| T \sum_{j=1}^{S_t} (w_1^l - w_1^j) \|$$
$$+ \| \sum_{k=1}^{T-1} \sum_{j=1}^{S_t} \alpha_{(T-k)} k \left( \nabla F(w_{(T-k)}^l) - \sum_{j=1}^{S_t} g^j(w_{(T-k)}^j, \xi_j) \right) \|$$
$$+ \| \sum_{k=1}^{T} \sum_{j=1}^{S_t} \sum_{q=k-\tau_k^j}^{q=k-1} (w_{q+1}^j - w_q^j) \| \Bigg) \tag{39}$$

$$\leq DL \| T \sum_{j=1}^{S_t} (w_1^l - w_1^j) \| + DL \cdot 2C^2 m^2 G \sum_{k=1}^{T-1} \alpha_{(T-k)} k$$
$$+ DLCmG \sum_{k=1}^{T} \sum_{q=k-\tau_k^j}^{q=k-1} \alpha_q \tag{40}$$

$$\leq DL \| T \sum_{j=1}^{S_t} (w_1^l - w_1^j) \| + 2C^2 m^2 GDL \sum_{k=1}^{T-1} \frac{\alpha k}{((1+\mu)T - k)^2}$$
$$+ DLCmG \sum_{k=1}^{T} \frac{\alpha \tau}{(k - \tau + T)^2} \tag{41}$$

$$< DL \| T \sum_{j=1}^{S_t} (w_1^l - w_1^j) \| + 2\alpha C^2 m^2 GDL \left( \frac{T}{\mu T + 1} + \ln |\mu T + 1| \right)$$
$$+ DLCmG\alpha\tau \left( \frac{1}{1 - \tau + T} \right) \tag{42}$$

Inequations 34 and 39 are from triangle inequality. Inequation 35 is from the fact $\nabla F(w_k^l) = \frac{1}{n} \sum_{i=1}^{n} \nabla F(w_k^l, \xi_i) = \sum_{j=1}^{m} g^j(w_k^l, \xi_j)$ and blockwise Lipschitz-continuity. Inequation 36 is from Assumption 1.1 and represents $\sum_{k=1}^{T} (\cdot)$ by a summand sequence.

Equation 37 comes from the fact

$$w_k^l - w_{k-\tau_k^j}^j = (w_k^l - w_k^j) + \sum_{q=k-\tau_k^j}^{q=k-1} (w_{q+1}^j - w_q^j)$$

Equation 38 comes from the fact

$$w_k^l - w_k^j = (w_1^l - \alpha_1 \nabla F(w_1^l) - ... - \alpha_k \nabla F(w_k^l))$$
$$- (w_1^j - \alpha_1 \sum_{j=1}^{S_t} g^j(w_1^j, \xi_j) - ... - \alpha_k \sum_{j=1}^{S_t} g^j(w_k^j, \xi_j))$$
$$= (w_1^l - w_1^j) - \alpha_1 (\nabla F(w_1^l) - \sum_{j=1}^{S_t} g^j(w_1^j, \xi_j))$$
$$- ... - \alpha_k (\nabla F(w_k^l) - \sum_{j=1}^{S_t} g^j(w_k^j, \xi_j)).$$

Inequation 40 follows Assumption 1.1 from which we obtain

$$\| \nabla F(w_{(T-k)}^l) - \sum_{j=1}^{S_t} g^j(w_{(T-k)}^j, \xi_j) \| \leq 2CmG,$$

and,

$$\| w_{q+1}^j - w_q^j \| = \| \alpha_q g^j(w_q^j, \xi_j) \| \leq \alpha_q G$$

Equation 41 is obtained by applying $\alpha_{(T-k)} = \frac{\alpha}{(T-k+\mu T)^2}$, and $\sum_{q=k-\tau_k^j}^{q=k-1} \alpha_q \leq \frac{\alpha\tau}{(k-\tau+T)^2}$.

Inequation 42 is from the following fact:
Since

$$\int_a^b \frac{k}{(c-k)^2} dk = \frac{b}{c-b} + \ln|c-b| - \frac{a}{c-a} - \ln|c-a|,$$

we have

$$\int_1^{T-1} \frac{k}{((1+\mu)T - k)^2} dk = \frac{T-1}{(1+\mu)T - (T-1)}$$
$$+ \ln|(1+\mu)T - (T-1)| - \frac{1}{(1+\mu)T - 1} - \ln|(1+\mu)T - 1|$$
$$< \frac{T-1}{(1+\mu)T - (T-1)} + \ln|(1+\mu)T - (T-1)| (\text{with } T \geq 2)$$
$$< \frac{T}{\mu T + 1} + \ln|\mu T + 1|$$

and

$$\int_1^T \frac{1}{(k - \tau + T)^2} dk = \frac{1}{1 - \tau + T} - \frac{1}{T - \tau + T}$$
$$< \frac{1}{1 - \tau + T}$$

$\square$

# G THE SECRECY OF $\widetilde{g}_k^l$

Recall that the involved parties are a set $\mathcal{L}$ of $m$ participants denoted with logical identities $l \in [1, m]$, and $t$ is the adversarial threshold ($t \leq m - 2$). Let $c$ be any subset of $\mathcal{L}$ that includes the compromised and colluding parties.

We demonstrate that, during optimization in CGD, given only the sum of the local gradients which are computed on different confined models, the adversary can learn no information other than their own inputs and the sum of the local gradients from other honest parties, i.e., $\sum \widetilde{g}_k^l(w_k^l, \xi_l)|_{l \in \mathcal{L} \setminus c}$.

Our analysis is based on the simulation paradigm [25]. It compares what an adversary can do in a real protocol execution to what it can do in an ideal scenario, which is secure by definition. The

adversary in the ideal scenario, is called the simulator. An indistinguishability between adversary's view in real and ideal scenarios guarantees that it can learn nothing more than their own inputs and the information required by the simulator for the simulation.

To facilitate the understanding on our analysis, we first present the used notations. Denote $g_k^{\mathcal{L}'} = \{\widetilde{g_k^l}(w_k^l, \xi_l)\}_{l \in \mathcal{L}'}$ as the local gradients of any subset of participants $\mathcal{L}' \subseteq \mathcal{L}$ at $k^{th}$ iteration. Let $VIEW_{real}(g_k^{\mathcal{L}}, t, \mathcal{P}, c)$ denote their combined views from the execution of a real protocol $\mathcal{P}$. Let $VIEW_{ideal}(g_k^c, z, t, \mathcal{F}_p, c)$ denote the views of $c$ from an ideal execution that securely computes a function $\mathcal{F}_p$, where $z$ is the information required by the simulator $\mathcal{S}$ in the ideal execution for simulation.

The following theorem shows that when executing CGD with the threshold $t$, the joint view of the participants in $c$ can be simulated by their own inputs and the sum of the local gradients from the remaining honest nodes, i.e., $\sum \widetilde{g_k^l} \mid_{l \in \mathcal{L} \setminus c}$. Therefore, $\sum \widetilde{g_k^l} \mid_{l \in \mathcal{L} \setminus c}$ is the only information that the adversary can learn during the execution.

THEOREM 3. *When executing CGD with the threshold $t$, there exists a simulator $\mathcal{S}$ such that for $\mathcal{L}$ and $c$, with $c \subseteq \mathcal{L}$ and $|c| \le t$, the output of $\mathcal{S}$ from $VIEW_{ideal}$ is perfectly indistinguishable from the output of $VIEW_{real}$, namely*

$$VIEW_{real}(g_k^{\mathcal{L}}, t, \mathcal{P}, c) \equiv VIEW_{ideal}(g_k^c, z, t, \mathcal{F}_p, c)$$

*where*

$$z = \sum \widetilde{g_k^l}(w_k^l, \xi_l) \mid_{l \in \mathcal{L} \setminus c} .$$

PROOF. We define $\mathcal{S}$ through each training iteration as:

$\textbf{SIM}_1$: $SIM_1$ is the simulator for the first training iteration.

Since the inputs of the parties in $c$ do not depend on the inputs of the honest parties in $\mathcal{L} \setminus c$, $SIM_1$ can produce a perfect simulation by running $c$ on their true inputs, and $\mathcal{L} \setminus c$ on a set of pseudorandom vectors $\eta_1^{\mathcal{L} \setminus c} = \{\eta_1^l\}_{l \in \mathcal{L} \setminus c}$ in a way that

$$\sum \eta_1^{\mathcal{L} \setminus c} = \sum \eta_1^l \mid_{l \in \mathcal{L} \setminus c} = \sum g_1^{\mathcal{L}} - \sum g_1^c = \sum g_1^l \mid_{l \in \mathcal{L} \setminus c} .$$

Since each $g_1^l(w_1^l, \xi_l)$ is computed from its respective confined model $w_1^l$ which is randomized in the initialization, the pseudorandom vectors $\eta_1^{m \setminus c}$ generated by $SIM_1$ for the inputs of all parties in $\mathcal{L} \setminus c$, and the joint view of $c$ in $VIEW_{ideal}$, will be identical to that in $VIEW_{real}$, namely

$$(\sum \eta_1^{\mathcal{L} \setminus c} + \sum g_1^c) \equiv \sum g_1^{\mathcal{L}},$$

and the information required by $SIM_1$ is $z = \sum g_1^l \mid_{l \in \mathcal{L} \setminus c}$.

$\textbf{SIM}_{k+1}(k \ge 1)$: $SIM_{k+1}$ is the simulator for the $(k+1)^{th}$ training iteration.

In *Real* execution, $\sum g_{k+1}^{\mathcal{L}}$ is computed as

$$\sum g_{k+1}^{\mathcal{L}} = \sum \widetilde{g_{k+1}^l}(w_{k+1}^l, \xi_l) \mid_{l \in \mathcal{L}} = \sum \widetilde{g_{k+1}^l}(w_k^l - \alpha_k \sum g_k^{\mathcal{L}}, \xi_l) \mid_{l \in \mathcal{L}}$$
$$= \sum \widetilde{g_{k+1}^l}(w_1^l - \sum_{i=1}^{k}(\alpha_i \sum g_i^{\mathcal{L}}), \xi_l) \mid_{l \in \mathcal{L}} . \quad (43)$$

In *Ideal* execution, since each $g_{k+1}^l(w_{k+1}^l, \xi_l) \mid_{l \in \mathcal{L}}$ is also computed from randomized $w_1^l$, $SIM_{k+1}$ can produce a perfect simulation by running the parties $\mathcal{L} \setminus c$ on a set of pseudorandom vectors $\eta_{k+1}^{\mathcal{L} \setminus c} = \{\eta_{k+1}^l\}_{l \in \mathcal{L} \setminus c}$ in a way that

$$\sum \eta_{k+1}^{\mathcal{L} \setminus c} = \sum \eta_{k+1}^l \mid_{l \in \mathcal{L} \setminus c} = \sum g_{k+1}^{\mathcal{L}} - \sum g_{k+1}^c = \sum \widetilde{g_{k+1}^l} \mid_{l \in \mathcal{L} \setminus c} .$$

As such, the joint view of $c$ in $VIEW_{ideal}$, will be identical to that in $VIEW_{real}$

$$(\sum \eta_{k+1}^{\mathcal{L} \setminus c} + \sum g_{k+1}^c) \equiv \sum g_{k+1}^{\mathcal{L}},$$

and the information required by $SIM_{k+1}$ is $z = \sum \widetilde{g_{k+1}^l} \mid_{l \in \mathcal{L} \setminus c}$.

By summarizing $SIM_1$ and $SIM_{k+1}$, the output of the simulator $VIEW_{ideal}$ of each training iteration is perfectly indistinguishable from the output of $VIEW_{real}$, and knowledge of $z$ is sufficient for the simulation.

□

## H BOUNDED ASYNCHRONY

Let $S_t$ denote a random set of $C \cdot m$ participants, and $\mathcal{A}$ denote the set of malicious participants, and $P_{\mathcal{A}}$ denote the malicious proportion, i.e., $P_{\mathcal{A}} = |\mathcal{A}|/m$. Given a participant $l$ at $k^{th}$ iteration, the probability of any participant $j$ being at $t^{th}$ iteration where $t \le k - \tau$ is

$$Pr(t \le k - \tau) \le Pr^\tau(S_t \cap \mathcal{A} \ne \emptyset) = (1 - Pr(S_t \cap \mathcal{A} = \emptyset))^\tau$$
$$= (1 - \binom{(1 - P_{\mathcal{A}})m}{Cm} / \binom{m}{Cm})^\tau \quad (\text{with } C < 1 - P_{\mathcal{A}}) \quad (44)$$

For example, with $P_{\mathcal{A}} = 0.2$ and $m = 30$, which is the commonly used setting in the existing FL literature [11, 18, 30], with $C = 0.2$, the probability of the staleness exceeding $\tau = 30$ is $1.33 \times 10^{-5}$, which is subpolynomially small.

## I DETAILS OF ABLATION STUDIES

### I.1 Scalability

In this section, we investigate how the number of participants affects CGD's privacy and accuracy performance. Figure 6 summarizes its performance with varying participant number. In general, CGD stays close to the centralized baseline in both validation loss and accuracy in all settings. As expected, CGD significantly outperforms the local training. For MNIST, in the worst case of $m = (1000 \times 112)$ when CGD contains the greatest number of participants, i.e., each participant owns a small proportion consisting of 60 samples with 7 features, CGD still achieves 0.191 and 95.78% in test loss and accuracy respectively, close to 0.081 and 97.54% of the centralized baseline. In contrast, the performance of local training declines to 2.348 and 11.64%. For CIFAR, the results on the test loss and accuracy are generally in line with that on the MNIST dataset. In the worst case of $m = (1000 \times 32)$, CGD achieves 1.271 and 67.67% in test loss and accuracy respectively, which significantly outperform 2.434 and 16.19% of the local training, and are close to the centralized baseline (0.675 and 75.72%). Table 5 and Table 6 list the detailed performance comparison with both centralized and local trainings. In all settings, the privacy bound $(\epsilon, \delta)$ of CGD remains lower than $(0.36, 10^{-5})$, and no significant difference is observed with varying number of participants. Table 7 shows the performance of CGD against active MIA with various number of participants. The results demonstrate that CGD effectively mitigates the threat, suppressing the attack accuracy around 50% in all settings. Table 8 shows the fairness performance of CGD on CIFAR with various number of participants. The standard deviation of test accuracy slightly increases with the growing number of participants, while retaining the superior performance. For example, CGD achieves 7.46% even with a large number of participants (i.e., 32,000

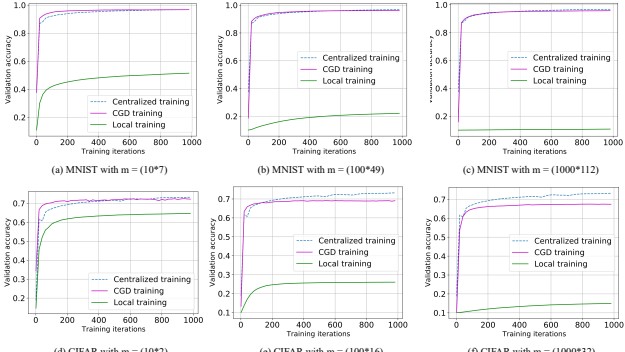

**(a) MNIST with m = (10*7)** **(b) MNIST with m = (100*49)** **(c) MNIST with m = (1000*112)**

**(d) CIFAR with m = (10*2)** **(e) CIFAR with m = (100*16)** **(f) CIFAR with m = (1000*32)**

**Figure 6: Results on the test accuracy for different number of participants in the honest-but-curious setting with privacy bounds $(\epsilon, \delta)$ at $(0.359, 10^{-5})$ on MNIST and $(0.350, 10^{-5})$ on CIFAR.**

**Table 5: Performance comparison on MNIST for various number of participants with CGD's privacy bounds $(\epsilon, \delta)$ at $(0.359, 10^{-5})$.**

| Training Methods | # of Participants | Test Loss | Test Accuracy |
|---|---|---|---|
| Centralized Training | N/A | 0.081 | 97.54% |
| Local Training | 10 × 7 | 1.283 | 54.39% |
| | 100 × 49 | 2.076 | 23.53% |
| | 1000 × 112 | 2.348 | 11.64% |
| CGD | 10 × 7 | 0.145 | 97.00% |
| | 100 × 49 | 0.164 | 96.19% |
| | 1000 × 112 | 0.191 | 95.78% |

**Table 6: Performance comparison on CIFAR for various number of participants with CGD's privacy bounds $(\epsilon, \delta)$ at $(0.350, 10^{-5})$.**

| Training Methods | # of Participants | Test Loss | Test Accuracy |
|---|---|---|---|
| Centralized Training | N/A | 0.675 | 75.72% |
| Local Training | 10 × 2 | 0.980 | 65.30% |
| | 100 × 16 | 2.008 | 26.25% |
| | 1000 × 32 | 2.434 | 16.19% |
| CGD | 10 × 2 | 0.821 | 72.45% |
| | 100 × 16 | 0.959 | 69.13% |
| | 1000 × 32 | 1.271 | 67.67% |

**Table 7: Performance of CGD against active MIA with various number of participants.**

| Datasets | # of Participants | Attack Accuracy |
|---|---|---|
| MNIST | 10 × 7 | 50.76% |
| | 100 × 49 | 51.46% |
| | 1000 × 112 | 50.57% |
| CIFAR | 10 × 2 | 50.00% |
| | 100 × 16 | 50.18% |
| | 1000 × 32 | 50.06% |

participants), outperforming Ditto [39] which achieves 12.23% with 16 participants (see Table 4).

**Table 8: Fairness performance of CGD with various number of participants on CIFAR.**

| # of Participants | Standard Deviation of Test Accuracy |
|---|---|
| 10 × 2 | 4.97% |
| 100 × 16 | 5.18% |
| 1000 × 32 | 7.46% |

**Table 9: Performance on validation accuracy with different initialization parameter $\lambda$.**

| Datasets | # of Participants | $\lambda = 0.1$ | $\lambda = 0.06$ | $\lambda = 0.01$ | $\lambda = 0.001$ | Random |
|---|---|---|---|---|---|---|
| MNIST | 1000 × 112 | 95.78% | 97.04% | 97.74% | 97.76% | 97.07% |
| CIFAR | 1000 × 32 | 67.59% | 70.17% | 73.99% | 69.43% | 71.01% |

## I.2 Effect of the Initialization

We conduct two experiments to investigate the effect of $\lambda$. We vary $\lambda$ while keeping other settings unchanging (*i.e.*, $\mu = 0$ and $T = 2000$). In the first experiment, we let all participants use the same $\lambda$ from 0.1, 0.06, 0.01, to 0.001, as they are around $\frac{1}{\sqrt{n}}$ (where $n$ is the sample size). In the other experiment, we let each participant randomly select its own $\lambda$ based on the uniform distribution within the range from 0.001 to 0.1.

The results of our first experiment are shown in Figure 7 and the first five columns in Table 9. In general, as $\lambda$ decreases, CGD achieves better performance. This confirms our expectation: decreasing $\lambda$ would reduce the value of $\| \mathop{\mathbb{E}}\limits_{j \in m^h} (w_1^l - w_1^j) \|$, such that the confined models are closer to the centralized optimum. We have not observed significant difference from $\lambda = 0.1$, 0.06 and 0.01. However, $\lambda$ cannot be set too small, in order to maintain the numerical stability in neural network [24]. For example, when we lower $\lambda$ to 0.001, the performance on CIFAR starts decreasing.

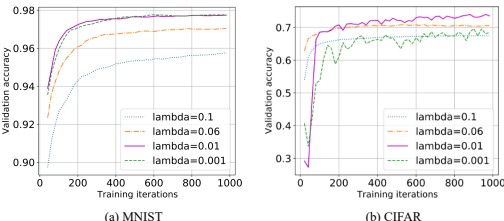

**(a) MNIST** **(b) CIFAR**

**Figure 7: Test accuracy for different $\lambda$ with $m = (1000 \times 112)$ on MNIST and $m = (1000 \times 32)$ on CIFAR.**

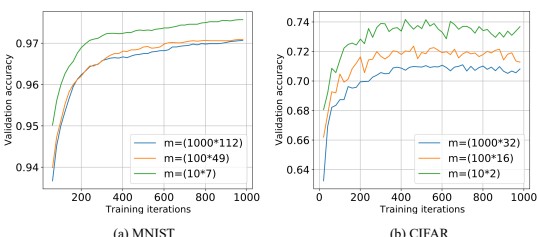

**(a) MNIST** **(b) CIFAR**

**Figure 8: Test accuracy with $\lambda$ selected uniformly at random.**

The results of our second experiment are shown in Figure 8 and the last column in Table 9. When the participants uniformly randomize their $\lambda$s from 0.001 to 0.1, the performance of CGD is still comparable with the centralized mode. For example, with $m = (1000 \times 112)$ participants on MNIST and $m = (1000 \times 32)$ on CIFAR, CGD achieves 97.07% and 71.01% in the test accuracy respectively. This suggests that CGD keeps robust when the $\lambda$s of its participants differ by two orders of magnitude.

In Appendix I.3, we provide additional results of the effect of the diminishment factor $\mu$.

## I.3   Effect of diminishment factor $\mu$

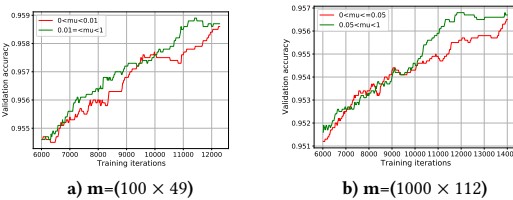

a) m=$(100 \times 49)$    b) m=$(1000 \times 112)$

**Figure 9: Test accuracy with different $\mu$ on the MNIST.**

In this experiment, we tune $\mu$ while keeping $\lambda$ fixed as 0.1. For the step size ($\alpha_k$), we run CGD with a fixed $\alpha$ till it (approximately) reaches a preferred point, and then continue our experiment with $\alpha_1 = \alpha$. This is to keep CGD practical, as each time the stepsize is diminished, more iterations are required. Therefore, the first 6,000 iterations is run with $\alpha = 0.01$, and then $\alpha_k$ decreases in each of the following 8,000 iterations.

Figure 9 demonstrates the results with varying $\mu$ for $m = (100 \times 49)$ and $m = (1000 \times 112)$. For the case of $m = (100 \times 49)$, $\mu \geq 0.01$ gives a slightly faster speed. CGD reaches the test accuracy of 95.9% at $11401^{th}$ iteration, 2,340 iterations (16.7% less) faster than training with $\mu < 0.01$. For the case of $m = 1000 \times 112$), $\mu > 0.05$ gives a faster speed. CGD reaches the test accuracy 95.68% at $11821^{th}$ iteration, 2,140 iterations (15.28% less) faster than training with $\mu \leq 0.05$. Even though such slight difference is observed, our experiment suggests that the effect of tuning $\mu$ is relatively limited. The gain of the test accuracy is only within 0.2% in the cost of around 2,000 iterations.

