# OpenReview forum: "Privacy-Preserving and Fairness-Aware Federated Learning for Critical Infrastructure Protection and Resilience"
_ACM.org/TheWebConf/2024/Conference — TheWebConf24_

### Official Review · Reviewer_GJ5E · 2023-11-17

**Novelty:** 2
**Technical Quality:** 3

**Review:**

1. The process of updating the global model after local pruning is unclear, particularly in terms of how heterogeneous models are aggregated.

1. Each client is required to select a certain proportion of updates for aggregation, which significantly increases communication overhead.

1. The claimed enhancement of privacy protection does not actually stem from pruning, but rather from the addition of noise defined by differential privacy. By reducing dimensions, sensitivity is lowered, but this is essentially no different from existing methods.

1. Furthermore, the facts that the author's idea relies on (lines 19-21) and Figure 2 are not convincing, lacking solid evidence or references to existing work. The summary in Table 1 overly downplays existing work and is not accurate.

1. The experiments lack comparison with the latest heterogeneous DFL aggregation mechanisms.

1. The background provided does not have a strong connection with the work described, and the scheme does not take into account the characteristics of industrial control equipment, despite the author's claim that the FL is designed for industrial equipment.

**Questions:**

Please see the review comment.

---

Rebuttals read.

**Reviewer Confidence:**

4: The reviewer is certain that the evaluation is correct and very familiar with the relevant literature

**Scope:**

3: The work is somewhat relevant to the Web and to the track, and is of narrow interest to a sub-community

---

### Official Review · Reviewer_WJ5t · 2023-11-22

**Novelty:** 6
**Technical Quality:** 6

**Review:**

This paper presents a novel serverless approach to federated learning, focusing on enhancing privacy and fairness. This design achieves better fairness and stronger privacy protection under the same level of noise. Overall, I believe the proposed framework is innovative and feasible, while with two concerns regarding the design choice and the experiments.

# Choice of using both MPC and DP

The design incorporates both Multi-Party Computation (MPC) and Differential Privacy (DP) for gradient privacy. While MPC secures gradient aggregation, DP introduces noise to local gradients.

Specifically, each participant generates local gradients in each training round and then adds noise to these local gradients (step 2 on page 5). Subsequently, a randomly selected subset of participants securely aggregates their gradients using MPC (step 3 on page 5). These aggregated gradients are then used to update the participants' local models, with the update of the local model being controlled by the Loss threshold LRR.

However, the concurrent use of both MPC and DP is confusing to me. Each method, independently, appears to focus on preventing the direct transmission of participants' gradients, thereby avoiding privacy leakage. MPC alone seems capable of ensuring secure gradient aggregation, which should sufficiently protect the participants' real inputs. Conversely, employing DP alone might also be feasible, where participants receive only gradients with noise from others. The overlapping roles of MPC and DP in the framework necessitates a clearer explanation of their combined utility in enhancing the overall privacy of the federated learning process.

# Efficiency of MPC

Concerning the computational cost of MPC, the authors note on page 3 that their approach "ensures the secrecy of individual updates while devolving the computational complexity to the local gradients." Given that cost is a common evaluation criterion in MPC-based solutions, I believe an analysis of this approach's computational cost should be included.

Furthermore, comparing this cost with that of traditional Federated Learning plus MPC approaches would enhance the evaluation. While I recognize that this paper introduces a decentralized solution, thereby distributing the computational cost across each participant's local server, a comparative analysis would nonetheless provide a clearer understanding of the approach's efficiency. This suggestion is somewhat nitpicking, and I would not press for these comparative evaluations, but their inclusion could enrich the paper's technical assessment, I expect.

**Questions:**

- Could you please elaborate on the specific reasons for using both MPC and DP in the proposed approach? How do these two methods complement each other, and what unique benefits does each method offer to the overall privacy of the federated learning process?

**Reviewer Confidence:**

3: The reviewer is confident but not certain that the evaluation is correct

**Scope:**

4: The work is relevant to the Web and to the track, and is of broad interest to the community

---

### Official Review · Reviewer_bB5a · 2023-11-23

**Novelty:** 5
**Technical Quality:** 6

**Review:**

- Overall, I like the proposed idea

- Motivation is a bit unclear: in the introduction, the authors describe that their approach aims to preserve the privacy of the data but it is unclear what is the threat from a potential privacy leak in the given example (i.e., critical infrastructure). I understand the example about fairness (lines 98-102) but I am missing a similar example with regards to privacy.

- Authors use a P2P network as a motivating example for their study assuming mobile users (i.e., electric vehicles) but they neither test nor provide any details on the behaviour of their approach in cases where there is high churn (users frequently become offline throughout the duration of the protocol). What is the impact of churn in the provided fairness and privacy?

- Apart from the accuracy, privacy and fairness evaluation, the system performance is quite brief (only section 6.1 actually). It would be great if authors could measure the performance of their prototype also in terms of latency

- Paper would benefit from a few more text passes so it is easier to read. e.g., very long sentences like lines 106-111 which constitute actually one sentence.

**Questions:**

- Page 5: “A subset of participants 𝑆 (randomly selected with the fraction parameter 𝐶)” → I am not sure I understand exactly how the participants are being selected and how the system protects itself from malicious attempts (e.g., to poison or disrupt the secure addition).

**Ethics Review Description:**

There are no ethical issues

**Reviewer Confidence:**

2: The reviewer is willing to defend the evaluation, but it is likely that the reviewer did not understand parts of the paper

**Scope:**

4: The work is relevant to the Web and to the track, and is of broad interest to the community

---

### Official Review · Reviewer_ZJHe · 2023-11-24

**Novelty:** 6
**Technical Quality:** 6

**Review:**

Federated learning (FL) enables multiple clients to cooperatively train a central machine learning model with the help of a central server. Nevertheless, in current FL frameworks, the global model parameters that need to be shared are still vulnerable to potential data leakage. In this paper, the authors introduce a new approach called Confined Gradient Descent (CGD) to bolster the privacy of FL. The authors support their proposal with both theoretical and empirical evidence to demonstrate its effectiveness.

Strengths:
1) The paper is well crafted and straightforward to follow.
2) Thorough experiments show the effectiveness of the proposed method.

Weaknesses:
1) The specific value of the loss threshold LRR is unspecified.
2) It's not evident whether the datasets adhere to an iid or non-iid distribution.

**Questions:**

1) The loss threshold LRR plays a pivotal role in the proposed method. However, the authors have not provided the exact LRR values for all datasets, and it is crucial to investigate the impact of varying LRR values.
2) An inherent characteristic of federated learning is the non-iid distribution of training data among clients. The authors should provide a clear explanation of how they simulate this non-iid setting for all datasets.
3) In Section 4.2, it's unclear whether the experiments are conducted in a synchronous or asynchronous setting. It would be advantageous to present the performance of the proposed method under both settings, given that Section 3.2 mentions CGD's asynchronous capability. If asynchronous setups are used, it is important to clearly outline how the authors simulate this asynchronous environment.

**Reviewer Confidence:**

4: The reviewer is certain that the evaluation is correct and very familiar with the relevant literature

**Scope:**

4: The work is relevant to the Web and to the track, and is of broad interest to the community

---

### Official Review · Reviewer_DUsh · 2023-12-04

**Novelty:** 3
**Technical Quality:** 3

**Review:**

Welcome to submit your work to WWW. The proposed CGD aims to improve fairness and privacy for federated learning by going towards a fully serverless manner. With a logical structure, the paper provides detailed discussions on privacy-preserving FL, fairness considerations in FL and threat models. The motivation is somehow unclear why such fully decentralized approach is necessary and what specific problems from WWW related applications/systems are calling for such design.

As to relevance to WWW community, it's unclear how CGD can contribute to Web services and systems. The evaluation of CGD is theoretic based, covering general ML models and datasets such as MNIST and CIFAR-10. The results in Section 4 and Section 5 are difficult to follow, especially as referring to figures and tables in appendix.

Overall, the fairness and privacy angles are valuable to explore but the proposed CGD design lacks convincing evidence for WWW services and systems.

**Questions:**

Given the concerns, there are one core questions for authors to ponder on:

- From the design angle, what's the core motivation for CGD to take such fully serverless path, especially for Web users and developers?

**Reviewer Confidence:**

3: The reviewer is confident but not certain that the evaluation is correct

**Scope:**

1: The work is irrelevant to the Web

---

### Decision · Program_Chairs · 2024-01-22

**Decision:**

Accept

**Comment:**

This paper presents a new serverless approach to federated learning, Confined Gradient Descent (CGD), which is focused on enhancing privacy and fairness. The reviewers are split on the quality of the paper, as several reviewers believes the work is innovative and practical, while others give more critical reviews.

 Overall, I feel the work is solid and the approach in interesting in the sense that it achieves better fairness and stronger privacy protection under the same level of noise.

 However, I do have one important concern: This paper proposes a privacy-preserving approach for FL, which is pretty general in my opinion. I don't see why the proposed method is specific to critical infrastructure protection, especially for energy sectors. So the first sentence in the abstract, "The energy industry is undergoing significant transformations as it strives to achieve net-zero emissions and future-proof its infrastructure, where every participant in the power grid has the potential to both consume and produce energy resources." seems pretty confusing to me.
 I suggest the authors change the paper title and modify the abstract if the paper is accepted.